

# Entanglement evolution and generalised hydrodynamics: interacting integrable systems

**Vincenzo Alba[1,2], Bruno Bertini[3⋆] and Maurizio Fagotti[4]**

**1** SISSA and INFN, via Bonomea 265, 34136, Trieste, Italy
**2** Institute for Theoretical Physics, Universiteit van Amsterdam,
Science Park 904, Postbus 94485, 1098 XH Amsterdam, The Netherlands
**3** Department of physics, FMF, University of Ljubljana,
Jadranska 19, SI-1000 Ljubljana, Slovenia
**4** LPTMS, CNRS, Univ. Paris-Sud, Université Paris-Saclay, 91405 Orsay, France

⋆ bruno.bertini@fmf.uni-lj.si

## Abstract

We investigate the dynamics of bipartite entanglement after the sudden junction of two leads in interacting integrable models. By combining the quasiparticle picture for the entanglement spreading with Generalised Hydrodynamics we derive an analytical prediction for the dynamics of the entanglement entropy between a finite subsystem and the rest. We find that the entanglement rate between the two leads depends only on the physics at the interface and differs from the rate of exchange of thermodynamic entropy. This contrasts with the behaviour in free or homogeneous interacting integrable systems, where the two rates coincide.

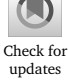
# 1 Introduction

Recent years witnessed interdisciplinary efforts aiming at understanding how statistical mechanics and thermodynamics arise from the out-of-equilibrium dynamics of isolated quantum many-body systems [1–5]. Characterising the entanglement spreading emerged as one of the key aspects to elucidate this issue [6–8]. The reason is twofold.

First, the spreading of entanglement provides universal information on the time evolution of the system, removing most of the inessential details that are typically tied to the correlation functions of local observables. This is best illustrated by considering the evolution of bipartite entanglement in pure states, customarily measured by the entanglement entropy [6–8]. Under mild hypotheses, preparing the system in a low entangled state and switching on spatially local interactions, the entanglement entropy of a finite subsystem exhibits a linear increase at intermediate times, whereas it saturates at asymptotically large times. This behaviour is observed in a huge variety of physical systems, ranging from random unitary circuits to integrable models [9–39]. In particular, the saturation value is extensive in the subsystem size and its density coincides with the density of the thermodynamic entropy of the statistical ensemble describing the steady state [40–47]. The latter is a Gibbs ensemble for generic systems and a Generalised Gibbs Ensemble (GGE) for integrable ones [2–4].

Second, the entanglement growth is crucial to understand the performance of numerical methods based on Matrix Product States, such as the time-dependent Density Matrix Renormalization Group (tDMRG) [48–54]. Specifically, the linear entanglement growth implies an exponential increase in time of the complexity of the numerical simulation. Accordingly, the value of the slope determines whether or not the simulation is feasible.

Despite its fundamental importance, exact results for the out-of-equilibrium dynamics of entanglement are typically extremely hard to obtain and, therefore, very scarce. Up to now they have been found only for models that can be mapped to free fermions [9], and, very recently, for a particular "maximally chaotic" Floquet system [38]. Nevertheless, two different effective descriptions have been put forward in the extreme cases of integrable [10] and chaotic [35–37] systems, which allow one to recover quantitatively the dynamics of entanglement entropies in the limit of large times and subsystem sizes, also called "space-time scaling limit". These descriptions are respectively known as "semiclassical quasiparticle picture" and

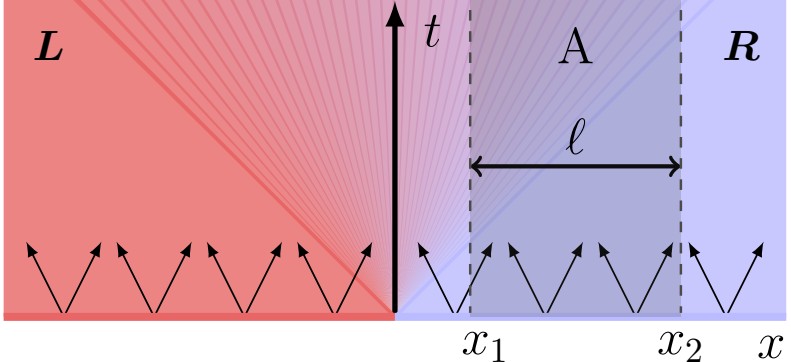

Figure 1: Sketch of the physical setting used in this work. At large times after the sudden junction of two macroscopically different states (left and right leads), local properties at fixed $x/t$ are described by a Local Quasi Stationary State (LQSS). Here $x$ is measured from the interface between the two chains. In this setting we are interested in the entanglement entropy of a region $A$ of length $\ell$. A semiclassical description in terms of free quasiparticles applies to the entanglement entropy. In the quasiparticle picture pairs of entangled quasiparticles are created in the bulk of the two chains. Quasiparticles forming an entangled pair have rapidity of opposite sign and initially travel with opposite velocities.

"minimal membrane picture". In this paper we are interested in the entanglement spreading in integrable models, thus we focus on the former.

In the semiclassical quasiparticle picture one views the initial state as a source of entangled pairs of quasiparticles, generated uniformly in space at the initial time and propagating as free classical objects with opposite velocities. Quasiparticles produced at the same point in space are mutually entangled, whereas quasiparticles created further apart are not. The entanglement between a subsystem $A$ and the rest is proportional to the total number of quasiparticles created at the same point and shared between $A$ and its complement at time $t$. Assuming that there are $N_s$ species of particles, whose dispersion relation is characterised by a real "rapidity" $\lambda$, we find

$$S_A(t) = \sum_{\alpha=1}^{N_s} \int d\lambda \int dx \, f_{\alpha,\lambda}(x) \chi_A[X_{\alpha,\lambda}(x,t)] \big(1 - \chi_A[X_{\alpha,-\lambda}(x,t)]\big), \tag{1}$$

where $\chi_A(x)$ is the characteristic function of the interval $A$, the function $f_{\alpha,\lambda}(x)$ denotes the contribution to the entanglement of the pair $(\alpha,\lambda),(\alpha,-\lambda)$, with rapidity $\pm\lambda$ and species $\alpha$, originated at $x$, and $X_{\alpha,\lambda}(x,t)$ gives the position at time $t$ of the quasiparticle that was created at position $x$ at time 0. This picture can be extended to the case where the initial state consists of more complicated multiplets of correlated particles [27, 28]. For the sake of simplicity, however, here we stick to cases where only pairs are produced and, moreover, we assume that the pairs are composed by particles of the same species.

The semiclassical quasiparticle picture, as described above, only gives qualitative predictions. To make it quantitatively accurate one needs to specify what are the semiclassical particles responsible for the entanglement spreading, what are their trajectories, and what is the contribution of a single entangled pair to the total entanglement. In translationally invariant interacting integrable systems, this task has been accomplished in Ref. [33]. This has been done by exploiting the exact knowledge of the stationary state describing finite subsystems at large times [3,4] to complement the semiclassical quasiparticle picture. In particular, Ref. [33] assumed that the entangling quasiparticles are the elementary excitations on the stationary

macrostate. This gives

$$f_{\alpha,\lambda}(x) = S_{\alpha,\lambda}^{YY}, \qquad X_{\alpha,\lambda}(x,t) = x + v_{\alpha,\lambda} t, \qquad (2)$$

where $v_{\alpha,\lambda}$ is the group velocity of the excitation $(\alpha, \lambda)$ and $S_{\alpha,\lambda}^{YY}$ is its contribution to the thermodynamic entropy. The quantitative prediction of Ref. [33] is conjectured to apply for all integrable models treatable via thermodynamic Bethe ansatz and to become exact in the space time scaling limit (at least for a class of "integrable" initial states [55]).

In this paper we derive an analogous quantitative prediction for cases where the initial state is not homogeneous. The idea is to use the recently developed theory of Generalised Hydrodynamics (GHD) [56, 57] (see Sec. 3) to determine the state of the system at large times and use it to complement the semiclassical quasiparticle picture. In particular, we focus on the paradigmatic case of initial states formed by the junction of two macroscopically different homogeneous states (leads) and determine the time evolution of the entanglement entropy of $A = [x_1, x_1 + \ell]$ (see Fig. 1) in the space-time scaling limit. Our prediction applies to generic integrable models and is tested in the concrete case of the anisotropic spin-1/2 Heisenberg chain.

Note that the entanglement spreading in inhomogeneous settings has been already investigated in the recent literature, see e.g. Refs. [58–63, 65, 66]. In particular, Refs. [61] and [65] considered exactly the problem studied here. Both these references, however, explored special cases. Ref. [65] investigated free systems, while Ref. [61] focussed on $x_1 = 0$ in the limits $\ell/t \to 0$ and $\ell/t \to \infty$. Our work represents a non-trivial extension of these studies.

Specifically, our prediction displays a remarkable novel effect due to the combination of inhomogeneity and interactions. This is most easily explained by considering the entropy of one of the leads, namely $A = [0, \infty[$, and comparing it with the homogeneous and the free cases. In particular, in the homogeneous case one has

$$S_{\text{lead}}(t) = t \sum_{\alpha=1}^{N_s} \int d\lambda \, |v_{\alpha,\lambda}| S_{\alpha,\lambda}^{YY} = t \sum_{\alpha=1}^{N_s} \int_{v_{\alpha,\lambda}>0} d\lambda \, v_{\alpha,\lambda} S_{\alpha,\lambda}^{YY} - t \sum_{\alpha=1}^{N_s} \int_{v_{\alpha,\lambda}<0} d\lambda \, v_{\alpha,\lambda} S_{\alpha,\lambda}^{YY}. \qquad (3)$$

In words: *the entanglement entropy increases with the rate at which the two leads exchange thermodynamic entropy*. Ref. [65] confirmed that, if the system is free, this feature remains true also in the inhomogeneous case. This is because, in both these cases, the trajectories of quasiparticles are straight lines; consequently, only one of the two entangled particles in a given pair can cross the junction, and one can forget about the pair structure. Here, instead, inhomogeneity and interactions cause the trajectories to curve, making it possible for both particles of a given entangled pair to cross the junction, and, in turn, slow down the rate of entanglement growth. In particular, this implies that the conjecture for the entanglement production rate put forward in Ref. [61] is generically incorrect (although the correction is typically small).

The rest of the manuscript is organised as follows. In Section 2 we review the Thermodynamic Bethe Ansatz treatment of generic integrable models. In Section 3 we summarise the GHD formalism for quenches from piecewise homogeneous initial states. In Section 4 we identify the entangling quasiparticles and present a detailed analysis of their trajectories. In Section 5 we derive and simplify the quasiparticle picture prediction for the full-time dynamics of the entanglement entropy. In Section 6 we analyse the entanglement production rate for the semi-infinite chain. In Section 7 we provide numerical checks on the validity of our results, by presenting tDMRG data for several quenches in the XXZ chain. Finally, in Section 8 we draw our conclusions. Three appendices contain various technical details of our derivations.

## 2 Thermodynamic Bethe Ansatz treatment

In this work we consider interacting integrable quantum many-body systems describable by Thermodynamic Bethe Ansatz [67] (TBA). This description applies to a variety of integrable models both in the continuum and on the lattice. In most of this paper we will keep the discussion at a general level, without specifying any concrete model. In Sec. 7 our results will be tested against numerical simulations in the paradigmatic example of the "gapped" XXZ spin-1/2 chain

$$H = \frac{J}{4} \sum_{j=1}^{L} \Big[ \sigma_j^x \sigma_{j+1}^x + \sigma_j^y \sigma_{j+1}^y + \Delta(\sigma_j^z \sigma_{j+1}^z - 1) \Big], \qquad \sigma_{L+1}^\alpha = \sigma_1^\alpha, \qquad (4)$$

where we denoted by $\sigma_j^\beta$ ($\beta = x, y, z$) the Pauli matrices at position $j$, by $L$ the volume of the system, and by $\Delta > 1$ the "anisotropy".

Let us now briefly summarise the main aspects of the TBA description that are needed in the rest of the paper. When a TBA description applies, in the thermodynamic limit $L \to \infty$ the eigenstates of the Hamiltonian are characterised by a set of functions

$$\{\rho_{\alpha,\lambda}\}_{\alpha=1,\dots,N_s}, \qquad (5)$$

where the real variable $\lambda \in C \subset \mathbb{R}$ is customarily called "rapidity" and the integer $N_s$ is the "number of species". The integer $N_s$, and the domain $C$ depend on the details of the specific model considered. For instance, in the XXZ chain with $\Delta > 1$ one has $N_s \to \infty$ and can choose $C = [-\pi/2, \pi/2]$.

The functions $\{\rho_{\alpha,\lambda}\}$ are known as "root densities", and they can be interpreted as rapidity distributions of the system's stable quasiparticles. The rapidity $\lambda$ parametrises the quasiparticles' dispersion relation while the index $\alpha$ labels different families. For instance, in the XXZ chain, $\alpha = 1$ corresponds to magnon-like excitations, whereas quasiparticles with $\alpha > 1$ can be thought of as bound states of $\alpha$ magnons. A quasiparticle of the species $\alpha$ with rapidity $\lambda$ has energy $e_\alpha(\lambda)$ and quasimomentum $p_\alpha(\lambda)$ given by [67]

$$p_\alpha(\lambda) = 2 \arctan\left(\frac{\tan(\lambda)}{\tanh(\alpha\eta/2)}\right), \qquad e_\alpha(\lambda) = \frac{-J \sinh\eta \sinh(\alpha\eta)}{\cosh(\alpha\eta) - \cos(2\lambda)}, \qquad (6)$$

where we introduced $\eta = \cosh^{-1}\Delta > 0$. In other words, the root densities generalise to interacting integrable models the notion of momentum occupation numbers in free systems.

The expectation values of local operators can be expressed as functionals of the root densities $\{\rho_{\alpha,\lambda}\}$. For instance, the densities $q_x$ of local conserved charges, i.e., operators with local densities commuting with the Hamiltonian, are expressed as simple linear functionals as follows

$$\langle\{\rho_{\alpha,\lambda}\}|q_x|\{\rho_{\alpha,\lambda}\}\rangle = \sum_{\beta=1}^{N_s} \int d\mu \, q_{\beta,\mu} \rho_{\beta,\mu}. \qquad (7)$$

Here the "bare charges" $q_{\beta,\mu}$ are functions specifying the charge density, and we denoted by $|\{\rho_{\alpha,\lambda}\}\rangle$ a generic eigenstate characterised by the set of root densities $\{\rho_{\alpha,\lambda}\}$.

The correspondence between eigenstates and root densities is generically not one-to-one: a large number of eigenstates corresponds to the same set of root densities. This fact is usually referred to by saying that the root densities specify a "thermodynamic macrostate" of the system, while the eigenstates of the Hamiltonian correspond to its "microstates". Quantitatively, for a finite system of size $L$ there are $\sim \exp[L \sum_\alpha \int d\lambda \, S_{\alpha,\lambda}^{YY}]$ eigenstates that in the limit $L \to \infty$ are described by the same set of densities $\{\rho_{\alpha,\lambda}\}$. Any of these eigenstates can be regarded as

a finite-size representative eigenstate of the thermodynamic macrostate. Here we introduced the Yang-Yang entropy density

$$S_{\alpha,\lambda}^{YY} \equiv S^{YY}[\rho_{\alpha,\lambda}] = -\rho_{\alpha,\lambda}^t \left[ \frac{\rho_{\alpha,\lambda}}{\rho_{\alpha,\lambda}^t} \log \frac{\rho_{\alpha,\lambda}}{\rho_{\alpha,\lambda}^t} + \left(1 - \frac{\rho_{\alpha,\lambda}}{\rho_{\alpha,\lambda}^t}\right) \log\left(1 - \frac{\rho_{\alpha,\lambda}}{\rho_{\alpha,\lambda}^t}\right) \right], \qquad (8)$$

where the auxiliary functionals $\{\rho_{\alpha,\lambda}^t\}$ called "total root densities" are defined as [67]

$$\rho_{\alpha,\lambda}^t = a_{\alpha,\lambda} + \sum_{\beta=1}^{N_s} \int \mathrm{d}\mu \, T_{\alpha,\lambda;\beta,\mu} \rho_{\beta,\mu}. \qquad (9)$$

The precise form of $a_{\alpha,\lambda}$ and $T_{\alpha,\lambda;\beta,\mu}$ depends, again, on the specific model considered. In general, however, the kernel $T_{\alpha,\lambda;\beta,\mu}$ encodes all the information about the two-particle scattering matrix (the only non trivial one in integrable models). For instance, for the gapped XXZ spin-1/2 chain we have [67]

$$a_{\alpha,\lambda} = \frac{1}{\pi} \frac{\sinh(\alpha\eta)}{\cosh(\alpha\eta) - \cos(2\lambda)}, \qquad (10)$$

$$T_{\alpha,\lambda;\beta,\mu} = (1 - \delta_{\alpha,\beta}) a_{|\alpha-\beta|,\lambda-\mu} + 2a_{|\alpha-\beta|+2,\lambda-\mu} + \cdots + 2a_{\alpha+\beta-2,\lambda-\mu} + a_{\alpha+\beta,\lambda-\mu}. \qquad (11)$$

The total root densities are interpreted as the densities of all possible values that the rapidities can take (the constraint originates from the fact that in finite volume the rapidities obey a set of non-trivial quantisation conditions [67]). These densities are generically not uniform and, as a consequence of the non-trivial interactions, depend on the root densities.

Before concluding, we note that the TBA description can be used also for some mixed states. This is true every time a generalised microcanonical representation applies [3, 5]. In other words, if the expectation values of local observables in the mixed state can be reproduced, in the thermodynamic limit, by expectation values on a single (carefully chosen) eigenstate of the Hamiltonian. For example, the TBA description can be used for systems in Gibbs and Generalised Gibbs states [3, 5].

## 3   GHD description of the local quasi-stationary state

After being initialised in an inhomogeneous state $\hat{\boldsymbol{\rho}}_0$ (see Fig. 1 for an example), the system performs a non-trivial time evolution, during which the expectation values of local observables display fast oscillations in both position $x$ and time $t$. At large times, however, these fast oscillations dephase away, and the expectation values become slow functions of $x$ and $t$. In this regime, it is reasonable to expect that the expectation values can be described by a quasi-stationary state $\hat{\boldsymbol{\rho}}_s(x,t)$ retaining some slow dependence on position and time. Namely, we expect

$$\mathrm{tr}\left[\mathcal{O}_x e^{-i\hat{H}t} \hat{\boldsymbol{\rho}}_0 e^{i\hat{H}t}\right] \overset{t \gg 1}{\sim} \mathrm{tr}\left[\mathcal{O}_x \hat{\boldsymbol{\rho}}_s(x,t)\right], \qquad (12)$$

where $H$ is the Hamiltonian of the system, $\mathcal{O}_x$ is a generic observable localised around the point $x$, and $\hat{\boldsymbol{\rho}}_s$ is the density matrix describing the quasi-stationary state. In these general terms, (12) can be interpreted as a hydrodynamic approximation; there are however limits where (12) becomes exact (as in the cases studied in this paper). In the context of quantum non-equilibrium dynamics the emergence of such a state was first proposed in Ref. [68], where it was called *locally quasi-stationary state* (LQSS). Specifically, based on the intuition developed for homogeneous quenches [2–5], it was argued that, at fixed $(x,t)$, the state $\hat{\boldsymbol{\rho}}_s(x,t)$ is a GGE constructed with the charges of the time evolving Hamiltonian. This means that $\hat{\boldsymbol{\rho}}_s(x,t)$

is homogeneous, stationary, and admits a "microcanonical" representation in terms of a TBA representative eigenstate, or, equivalently, of a set of root densities $\{\rho_{\alpha,\lambda}(x,t)\}$. Note that the densities depend on space and time.

Determining $\hat{\boldsymbol{\rho}}_s(x,t)$ without solving the full non-equilibrium dynamics (in the presence of integrable interactions) is the key result of the theory of generalised hydrodynamics (GHD) introduced in Refs. [56, 57]. Specifically it was shown that, at the leading order in $x$ and $t$, the position-dependent root densities fulfil the following continuity equation

$$\partial_t \rho_{\alpha,\lambda}(x,t) + \partial_x(v_{\alpha,\lambda}(x,t)\rho_{\alpha,\lambda}(x,t)) = 0. \tag{13}$$

The quantity $v_{\alpha,\lambda}(x,t)$ appearing in (13) is the velocity of the elementary excitations on the state described by $\{\rho_{\alpha,\lambda}(x,t)\}$, see Ref. [69], and it is defined through the following integral equation

$$v_{\alpha,\lambda}(x,t)\rho_{\alpha,\lambda}^t(x,t) = v_{\alpha,\lambda}^{\mathrm{b}} a_{\alpha,\lambda} + \sum_{\beta=1}^{N_s} \int \mathrm{d}\mu\, T_{\alpha,\lambda;\beta,\mu}\, v_{\beta,\mu}(x,t)\rho_{\beta,\mu}(x,t), \tag{14}$$

where $\rho_{\alpha,\lambda}^t(x,t)$ is the $(x,t)$-dependent total root density (cf. Eq. (9))

$$\rho_{\alpha,\lambda}^t(x,t) = a_{\alpha,\lambda} + \sum_{\beta=1}^{N_s} \int \mathrm{d}\mu\, T_{\alpha,\lambda;\beta,\mu}\, \rho_{\beta,\mu}(x,t). \tag{15}$$

The model-dependent function $v_{\alpha,\lambda}^{\mathrm{b}}$ is the velocity of the excitations on the "vacuum state" (the state with $\rho_{\alpha,\lambda} = 0$) and is known as "bare velocity". For the gapped XXZ spin-1/2 chain we have

$$v_{\alpha,\lambda}^{\mathrm{b}} = -J \frac{\sinh\eta}{2} \frac{a_{\alpha,\lambda}'}{a_{\alpha,\lambda}}, \tag{16}$$

where $a_{\alpha,\lambda}$ is given in Eq. (11). Note that for interacting models, i.e. when $T_{\alpha,\lambda;\beta,\mu} \neq 0$, the velocity $v_{\alpha,\lambda}(x,t)$ depends on the densities $\{\rho_{\alpha,\lambda}(x,t)\}$. This makes equation (13) highly non-trivial.

The simplification introduced by Eq. (13) is remarkable. To determine the late-time properties of an integrable quantum many-body system one needs to solve a system of differential equations whose number is proportional to the system size, instead of solving the Schrödinger equation, which has instead the dimension of the Hilbert space. There is, however, a remaining non-trivial step to make before a solution can be obtained: one has to impose the initial conditions for $\rho_{\alpha,\lambda}(x,t)$. This has been successfully done in a number of cases [56,57,70–94], including the bipartite quench protocol considered here (see below), but in general the problem is still open. We note that, very recently, it has been shown that GHD provides the precise framework to describe experiments with trapped cold atoms [70].

Equation (13) admits a very simple interpretation in terms of a "kinetic theory" of free classical particles moving in an inhomogeneous background. One regards $\rho_{\alpha,\lambda}(x,t)$ as the distribution function for classical particles of the species $\alpha = 1,\ldots,N_s$ with momentum $\lambda$, at position $x$, and at time $t$. Equation (13) describes the evolution of the distribution functions $\rho_{\alpha,\lambda}$ due to the motion of the particles, which are nothing but a "coarse grained version" of the stable *interacting* quasiparticles characterising integrable models. Indeed, at the leading order in $(x,t)$, the only effect of the interaction is a renormalisation of the group velocity $v_{\alpha,\lambda}(x,t)$. Note that, at sub-leading orders, the effect of interactions might spoil this interpretation [93–96].

Finally, it is convenient to observe that, given a set of quantities $\{g_{\alpha,\lambda}(x,t)\}$ fulfilling (13), we have

$$\partial_t\left(\frac{g_{\alpha,\lambda}(x,t)}{\rho_{\alpha,\lambda}^t(x,t)}\right) + v_{\alpha,\lambda}(x,t)\partial_x\left(\frac{g_{\alpha,\lambda}(x,t)}{\rho_{\alpha,\lambda}^t(x,t)}\right) = 0. \tag{17}$$

Equation (17) has a material-derivative form and it is generically easier to solve than (13), for instance by using the method of characteristics [74, 75]. This implies that, instead of solving (13), it is convenient to define the "so-called" filling functions

$$\vartheta_{\alpha,\lambda}(x,t) \equiv \frac{\rho_{\alpha,\lambda}(x,t)}{\rho^t_{\alpha,\lambda}(x,t)}. \tag{18}$$

For the upcoming analysis it also important to note that the $(x,t)$-dependent Yang-Yang entropy densities

$$S^{YY}_{\alpha,\lambda}(x,t) = -\rho^t_{\alpha,\lambda}(x,t)\Big[\vartheta_{\alpha,\lambda}(x,t)\log\vartheta_{\alpha,\lambda}(x,t) + \big(1-\vartheta_{\alpha,\lambda}(x,t)\big)\log\big(1-\vartheta_{\alpha,\lambda}(x,t)\big)\Big] \tag{19}$$

fulfil a continuity equation of the form (13)

$$\partial_t S^{YY}_{\alpha,\lambda}(x,t) + \partial_x(v_{\alpha,\lambda}(x,t)S^{YY}_{\alpha,\lambda}(x,t)) = 0, \tag{20}$$

as it readily follows from (13) and (17).

## 3.1 Bipartite quench

Let us now specialise the GHD formalism of the previous section to what will be referred to as a "bipartite quench" (see Fig. 1). This is the time evolution of a state that, up to irrelevant[1] corrections localised around the junction, has the form

$$\hat{\rho}_0 \sim \hat{\rho}_{\mathrm{L}} \otimes \hat{\rho}_{\mathrm{R}}, \tag{21}$$

with $\hat{\rho}_{\mathrm{L(R)}}$ two macroscopically different homogeneous states. For instance, in our numerical tests in the XXZ chain we consider

$$\hat{\rho}_{\mathrm{L(R)}} \in \{|\mathrm{D}\rangle\langle\mathrm{D}|, |\mathrm{F},\theta\rangle\langle\mathrm{F},\theta|, |\mathrm{N},\theta\rangle\langle\mathrm{N},\theta|\}, \tag{22}$$

where we defined the "Dimer state" $|\mathrm{D}\rangle$, the "tilted Néel" state $|\mathrm{N},\theta\rangle$, and the "tilted ferromagnetic state" $|\mathrm{F},\theta\rangle$ as follows

$$|\mathrm{D}\rangle = \bigotimes_i \left(\frac{|\uparrow\downarrow\rangle - |\downarrow\uparrow\rangle}{\sqrt{2}}\right), \tag{23}$$

$$|\mathrm{N},\theta\rangle = e^{i\frac{\theta}{2}\sum_j \sigma^x_j}\frac{1}{\sqrt{2}}\Big[|\uparrow\downarrow\cdots\uparrow\downarrow\rangle + |\downarrow\uparrow\cdots\downarrow\uparrow\rangle\Big], \tag{24}$$

$$|\mathrm{F},\theta\rangle = e^{i\frac{\theta}{2}\sum_j \sigma^x_j}|\uparrow\uparrow\uparrow\ldots\rangle. \tag{25}$$

In this setting, if $\hat{\rho}_{\mathrm{L(R)}}$ have cluster decomposition properties, at long enough times all the quantities generally become functions of the "ray" $\zeta = x/t$ [56, 57, 68]; in the limit $t \to \infty$ the LQSS on each ray becomes exactly stationary, and all the sub-leading corrections to (13) vanish. Rewriting (17) (for $\vartheta_{\alpha,\lambda}(\zeta)$) in the variable $\zeta$ we have

$$\big(\zeta - v_{\alpha,\lambda}(\zeta)\big)\partial_\zeta\vartheta_{\alpha,\lambda}(\zeta) = 0. \tag{26}$$

In this case, whenever information propagates with a bounded velocity, it is possible to impose the initial conditions in (26) at $\zeta \to \pm\infty$ and solve it [56, 57, 68]. Indeed, at infinite distances from the interface between the two leads there are regions where no information on the inhomogeneity can arrive, and local observables evolve as if the system were homogeneous. This

---

[1]A quasi-localised impurity in the initial state does not change the late time behaviour if all the excitations are delocalised.

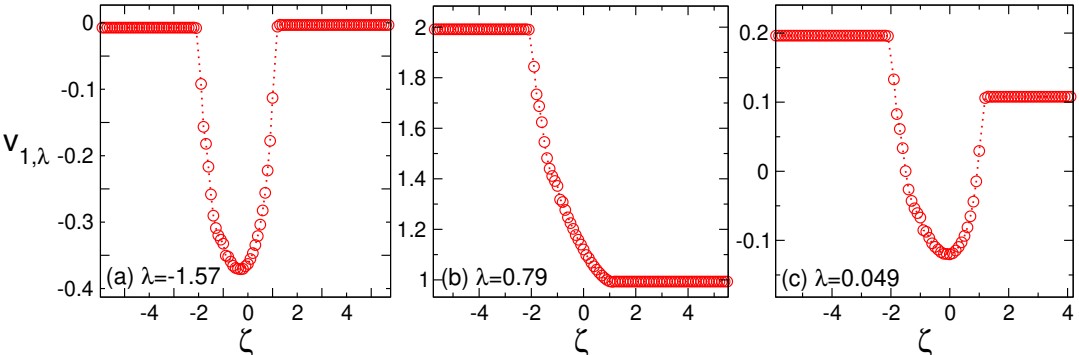

Figure 2: Velocity field $v_{\alpha,\lambda}$ after a quench in the XXZ chain for $\Delta = 10$, plotted against $\zeta \equiv x/t$. The pre-quench initial state is obtained by joining the Néel state $|N,0\rangle$ (24) (left) and the ferromagnet $|F,0\rangle$ (25) (right). Here we only show results for $\alpha = 1$. Different panels correspond to different values of rapidity $\lambda$. Note that for $\zeta < v^{\min} \approx -2$ and $\zeta > v^{\max} \approx 1$ the velocity field does not depend on $\zeta$, as expected.

means that their expectation values are described by stationary states that can be computed using the standard techniques developed for homogeneous quenches [2–5]. These stationary states provide the boundary conditions $\vartheta_{\alpha,\lambda}^{(R/L)}$ for (26). We then find

$$\vartheta_{\alpha,\lambda}(\zeta) = \vartheta_{\alpha,\lambda}^{(L)}\,\theta_H(v_{\alpha,\lambda}(\zeta) - \zeta) + \vartheta_{\alpha,\lambda}^{(R)}\,\theta_H(\zeta - v_{\alpha,\lambda}(\zeta)), \tag{27}$$

where $\theta_H(x)$ is the step function. Equation (27) is only an implicit solution because it depends on the velocity $v_{\alpha,\lambda}(\zeta)$ that in turn depends on $\vartheta_{\alpha,\lambda}(\zeta)$. To find $\vartheta_{\alpha,\lambda}(\zeta)$, it is convenient to adopt an iterative approach, combining (27) with the infinite time limit of (14) and (15). Namely

$$\rho_{\alpha,\lambda}^t(\zeta) = a_{\alpha,\lambda} + \sum_{\beta=1}^{N_s} \int d\mu\, T_{\alpha,\lambda;\beta,\mu}\,\vartheta_{\alpha,\lambda}(\zeta)\rho_{\beta,\mu}^t(\zeta), \tag{28}$$

$$v_{\alpha,\lambda}(\zeta)\rho_{\alpha,\lambda}^t(\zeta) = v_{\alpha,\lambda}^b a_{\alpha,\lambda} + \sum_{\beta=1}^{N_s} \int d\mu\, T_{\alpha,\lambda;\beta,\mu}\,\vartheta_{\alpha,\lambda}(\zeta)v_{\beta,\mu}(\zeta)\rho_{\beta,\mu}^t(\zeta). \tag{29}$$

Finally, for the upcoming discussion it is useful to mention that the continuity equation for the Yang-Yang entropy in terms of $\zeta$ reads as

$$\zeta\partial_\zeta S_{\alpha,\lambda}^{YY}(\zeta) - \partial_\zeta(v_{\alpha,\lambda}(\zeta)S_{\alpha,\lambda}^{YY}(\zeta)) = 0. \tag{30}$$

## 4 Entangling quasiparticles in inhomogeneous backgrounds

The first step to turn quantitative the quasiparticle picture for the entanglement dynamics is to identify the entangling quasiparticles and determine their trajectories. Here we perform this task in the case of interacting integrable models after bipartite quenches.

The basic assumption of this work is that the entangling quasiparticles are the excitations on the locally quasi-stationary state. The logic of this assumption is that, since the stable quasiparticle excitations in integrable models are responsible for the spreading of any kind of information, it is natural to argue that they also spread the entanglement. An analogous assumption has been formulated in Ref. [33] in the homogeneous case and in Ref. [65] for free systems.

To be specific, we denote by $X_{\alpha,\lambda}(x,t)$ the position at time $t$ of a particle of species $\alpha = 1,\ldots,N_s$ with rapidity $\lambda$ that started at position $x$. The position is measured from the interface between the two chains (see Fig. 1). We then assume that $X_{\alpha,\lambda}(x,t)$ is determined by the following classical equation of motion

$$\frac{\mathrm{d}}{\mathrm{d}t}X_{\alpha,\lambda}(x,t) = v_{\alpha,\lambda}\left(X_{\alpha,\lambda}(x,t),t\right), \tag{31}$$

where $v_{\alpha,\lambda}(x,t)$ are the *same velocities* as in (13). Note that the trajectories generated by (31) are generically not straight, in contrast to the homogeneous [33] and free [65] cases.

For bipartite quenches (see Fig. 1), $v_{\alpha,\lambda}(x,t)$ are obtained by solving the GHD equations (27)–(29) and are functions of the ratio $\zeta = x/t$, i.e.,

$$v_{\alpha,\lambda}(x,t) = v_{\alpha,\lambda}(x/t). \tag{32}$$

The function $v_{\alpha,\lambda}(\zeta)$ is taken to be fairly generic; we only make one assumption on its $\zeta$ dependence:

**Assumption 1.** *For fixed $\lambda$ and $\alpha$, the equation $\zeta - v_{\alpha,\lambda}(\zeta) = 0$ has a unique solution, which we call $\zeta_{\alpha,\lambda}$.*

This is the standard assumption of GHD. It has been verified in all the examples examined up to now, but it has not yet been proven rigorously. A number of interesting properties of the velocity field follow from (27)–(29). For example one can show (see Appendix A)

(i) $v_{\alpha,\lambda}(\zeta) = \begin{cases} v_{\alpha,\lambda}(-\infty) & \zeta < \min_\alpha \min_\lambda (v_{\alpha,\lambda}(-\infty)) \equiv v^{\min}, \\ v_{\alpha,\lambda}(\infty) & \zeta > \max_\alpha \max_\lambda (v_{\alpha,\lambda}(\infty)) \equiv v^{\max}, \end{cases}$

(ii) $\partial_\zeta v_{\alpha,\lambda}(\zeta)$ is bounded for all $\zeta$.

The first property stems from the bounds on the velocity at which information propagates from the interface to the bulk of the two semi-infinite chains; in space-time regions outside the lightcone spreading from the origin, the system is described by the macrostates $\rho_{\alpha,\lambda}^{(\mathrm{L/R})}$. A concrete example of a $\zeta$-dependent velocity field obtained in the XXZ model is reported in Fig. 2.

We remark that the quasiparticle picture is based on coarse graining procedure in space, momentum, and, in turn, time, so the coordinates $x$ and $t$ in (31) have some intrinsic indetermination. To keep track of potential effects of that, we naively capture such corrections with a single parameter $t_0$, which is regarded as the time when the initial conditions for the classical problem (31) are imposed

$$X_{\alpha,\lambda}(x,t_0) = x. \tag{33}$$

Keeping a finite $t_0 > 0$ regularises the initial value problem (31)(33). Indeed, properties (i) and (ii) ensure that $v_{\alpha,\lambda}(x/t)$ is a Lipschitz continuous function of $x$ for all $t > 0$. This implies that Cauchy's Theorem applies and a unique solution exists for any given initial condition $x$. Eventually we will take the limit $t_0/t \to 0$.

An immediate consequence of the uniqueness of the solution of (31)(33) is that, for fixed $(\alpha,\lambda)$, trajectories with different initial conditions cannot cross. In particular, it is useful to observe that quasiparticles with "initial velocity" $\zeta_{\alpha,\lambda}$ (*cf.* Assumption 1) follow linear trajectories, i.e., $s_{\alpha,\lambda}(t) = \zeta_{\alpha,\lambda}t$ solves (31) with initial value $\zeta_{\alpha,\lambda}t_0$. This implies

$$\begin{aligned} x > \zeta_{\alpha,\lambda}t_0 & \quad\Longleftrightarrow\quad & X_{\alpha,\lambda}(x,t) > \zeta_{\alpha,\lambda}t & \quad \forall t > t_0, \\ x < \zeta_{\alpha,\lambda}t_0 & \quad\Longleftrightarrow\quad & X_{\alpha,\lambda}(x,t) < \zeta_{\alpha,\lambda}t & \quad \forall t > t_0. \end{aligned} \tag{34}$$

Equivalently, this means that a trajectory starting on the left or on the right of $s_{\alpha,\lambda}(t)$ remains as such at any time.

To gain information on the qualitative form of a generic trajectory $X_{\alpha,\lambda}(x,t)$, it is useful to explicitly integrate Eq. (31). This is done by means of the following convenient rewriting

$$\frac{1}{t} = \frac{\frac{\mathrm{d}}{\mathrm{d}t}\left(X_{\alpha,\lambda}(x,t)/t\right)}{v_{\alpha,\lambda}(X_{\alpha,\lambda}(x,t)/t) - X_{\alpha,\lambda}(x,t)/t}. \tag{35}$$

Integrating the expression (35) from $t_0$ to $t$ we obtain

$$\int_{x/t_0}^{X_{\alpha,\lambda}(x,t)/t} \frac{\mathrm{d}\zeta}{v_{\alpha,\lambda}(\zeta) - \zeta} = \log\frac{t}{t_0}. \tag{36}$$

After taking the limit $t \to \infty$, the right-hand-side diverges logarithmically. The only way for the left-hand-side to match this behaviour is by having

$$\lim_{t\to\infty} \frac{X_{\alpha,\lambda}(x,t)}{t} = \zeta_{\alpha,\lambda}. \tag{37}$$

In words, this means that *the trajectories of the quasiparticles become linear at asymptotically long times*.

Crucially, Equation (36) allows us to explicitly determine the initial condition $x$ in terms of the position of the quasiparticle at time $t$. After a straightforward derivation (see Appendix A), we find

$$\frac{x}{t} = \theta_H\left(\zeta_{\alpha,\lambda} - \frac{X_{\alpha,\lambda}(x,t)}{t}\right)[v^{\min} - v_{\alpha,\lambda}(-\infty)]\exp\left[\int_{v^{\min}}^{X_{\alpha,\lambda}(x,t)/t} \frac{\mathrm{d}z}{z - v_{\alpha,\lambda}(z)}\right]$$
$$+ \theta_H\left(\frac{X_{\alpha,\lambda}(x,t)}{t} - \zeta_{\alpha,\lambda}\right)[v^{\max} - v_{\alpha,\lambda}(\infty)]\exp\left[\int_{X_{\alpha,\lambda}(x,t)/t}^{v^{\max}} \frac{\mathrm{d}z}{v_{\alpha,\lambda}(z) - z}\right] + O\left(\frac{t_0}{t}\right). \tag{38}$$

Here the first and the second terms account for quasiparticles created on the left and right lead (see Fig. 1), respectively. To treat the large time limit it is useful to define the scaling function

$$\Phi_{\alpha,\lambda}(\zeta, t, t_0) = \frac{X_{\alpha,\lambda}(\zeta t, t)}{t}, \tag{39}$$

such that $\Phi_{\alpha,\lambda}(\zeta, t, t_0)t$ is the position at time $t$ of the particle that was originated at position $\zeta t$ at time $t_0$. From (38) it is straightforward to find

$$\zeta = \theta_H(\zeta_{\alpha,\lambda} - \Phi_{\alpha,\lambda}(\zeta, t, t_0))[v^{\min} - v_{\alpha,\lambda}(-\infty)]\exp\left[\int_{v^{\min}}^{\Phi_{\alpha,\lambda}(\zeta,t,t_0)} \frac{\mathrm{d}z}{z - v_{\alpha,\lambda}(z)}\right]$$
$$+ \theta_H(\Phi_{\alpha,\lambda}(\zeta, t, t_0) - \zeta_{\alpha,\lambda})[v^{\max} - v_{\alpha,\lambda}(\infty)]\exp\left[\int_{\Phi_{\alpha,\lambda}(\zeta,t,t_0)}^{v^{\max}} \frac{\mathrm{d}z}{v_{\alpha,\lambda}(z) - z}\right] + O\left(\frac{t_0}{t}\right). \tag{40}$$

Furthermore, it is useful to introduce the inverse function $Z_{\alpha,\lambda}(\phi, t, t_0) = \Phi_{\alpha,\lambda}^{-1}(\phi, t, t_0)$. $Z_{\alpha,\lambda}(\phi, t, t_0)t$ gives the position at time $t_0$ of the particle $(\alpha, \lambda)$ that at time $t$ has position $\phi t$. Using (40) we obtain

$$Z_{\alpha,\lambda}(\phi, t, t_0) = \theta_H(\zeta_{\alpha,\lambda} - \phi)[v^{\min} - v_{\alpha,\lambda}(-\infty)]\exp\left[\int_{v^{\min}}^{\phi} \frac{\mathrm{d}z}{z - v_{\alpha,\lambda}(z)}\right]$$
$$+ \theta_H(\phi - \zeta_{\alpha,\lambda})[v^{\max} - v_{\alpha,\lambda}(\infty)]\exp\left[\int_{\phi}^{v^{\max}} \frac{\mathrm{d}z}{v_{\alpha,\lambda}(z) - z}\right] + O\left(\frac{t_0}{t}\right). \tag{41}$$

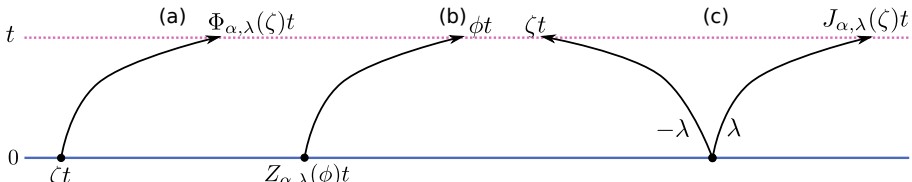

Figure 3: Pictorial representation of the functions $\Phi_{\alpha,\lambda}(\zeta)$ (*cf.* (40)), its inverse $Z_{\alpha,\lambda}(\phi)$ (*cf.* (41)), and of $J_{\alpha,\lambda}(\zeta)$ (*cf.* (63)). In (a) a quasiparticle is produced at time 0 at position $\zeta t$. The function $\Phi_{\alpha,\lambda}(\zeta)$ gives its ray at time $t$. In (b) $Z_{\alpha,\lambda}(\phi)t$ is the position at time 0 of the particle that at time $t$ is on the ray $\phi$. In (c) two quasiparticles with opposite rapidities $\lambda$ and $-\lambda$ are created. The function $J_{\alpha,\lambda}(\zeta)$ gives the ray at time $t$ of the quasiparticle whose partner with rapidity $-\lambda$ is at ray $\zeta$.

From now on we take the limit $t_0/t \to 0$, dropping the dependence on $t_0$ and $t$ in $\Phi_{\alpha,\lambda}$ and $Z_{\alpha,\lambda}$. With a slight abuse of notations, we indicate by $\Phi_{\alpha,\lambda}(\zeta)t$ the position at time $t$ of the quasiparticle that started at position $\zeta t$ at time 0 and by $Z_{\alpha,\lambda}(\phi)t$ the position at time 0 of the quasiparticle that is at position $\phi t$ at time $t$. The functions $\Phi_{\alpha,\lambda}$ and $Z_{\alpha,\lambda}$ are pictorially illustrated in Fig. 3. Using these definitions, one immediately has

$$Z_{\alpha,\lambda}(\Phi_{\alpha,\lambda}(\zeta)) = \zeta. \tag{42}$$

We mention that the function $Z_{\alpha,\lambda}(\phi)t$ has been already introduced in Refs. [74,75], where it is called "characteristics".

An important property of $Z_{\alpha,\lambda}(\phi)$ and $\Phi_{\alpha,\lambda}(\zeta)$ is that they are monotonic functions of their arguments. This is a direct consequence of the non-crossing condition (34) of the trajectories. To show the monotonicity of $Z_{\alpha,\lambda}(\phi)$ it is useful to observe that (41) implies

$$(\phi - v_{\alpha,\lambda}(\phi))\partial_\phi Z_{\alpha,\lambda}(\phi) = Z_{\alpha,\lambda}(\phi). \tag{43}$$

Moreover, a trivial consequence of (34) is

$$\frac{Z_{\alpha,\lambda}(\phi)}{\phi - v_{\alpha,\lambda}(\phi)} > 0 \qquad \forall \phi. \tag{44}$$

Equation (43) and (44) imply that $Z_{\alpha,\lambda}$ and, in turn, its inverse $\Phi_{\alpha,\lambda}(\zeta)$ are monotonic.

## 4.1 Flow of conserved quantities

In this section we show how the information about the quasiparticles' trajectories can be used to determine the flow of conserved quantities. The latter are represented by functions of $x$ and $t$ written in terms of a single-particle contribution $g_{\alpha,\lambda}(x,t)$ that satisfies the continuity equation (13), namely

$$G(x,t) = \sum_{\alpha=1}^{N_s} \int d\lambda \, g_{\alpha,\lambda}(x,t), \tag{45}$$

where

$$\partial_t g_{\alpha,\lambda}(x,t) + \partial_x v_{\alpha,\lambda}(x,t) g_{\alpha,\lambda}(x,t) = 0. \tag{46}$$

This definition includes the expectation values of all local and quasilocal conserved-charge densities, for which (*cf.* (7))

$$g_{\alpha,\lambda}(x,t) = q_{\alpha,\lambda} \rho_{\alpha,\lambda}(x,t), \tag{47}$$

but also the Yang-Yang entropy, as demonstrated by Eq. (20).

Since $g_{\alpha,\lambda}(x,t)$ is conserved, we have

$$\int_{X_{\alpha,\lambda}(a,t)}^{X_{\alpha,\lambda}(x,t)} \mathrm{d}y \; g_{\alpha,\lambda}(y,t) = \int_{a}^{x} \mathrm{d}y \; g_{\alpha,\lambda}(y,0). \tag{48}$$

Let us now specialise this relation to our setting

$$g_{\alpha,\lambda}(x,0) = \theta_H(x)g_{\alpha,\lambda}^{(R)} + \theta_H(-x)g_{\alpha,\lambda}^{(L)} \tag{49}$$

and assume that all the functions of $x$ and $t$ become functions only of the "ray" $x/t$. In this case (48) becomes

$$\int_{\zeta_{\alpha,\lambda}}^{\Phi_{\alpha,\lambda}(\zeta)} \mathrm{d}y \; g_{\alpha,\lambda}(y) = \zeta \left( \theta_H(\zeta)g_{\alpha,\lambda}^{(R)} + \theta_H(-\zeta)g_{\alpha,\lambda}^{(L)} \right), \tag{50}$$

where we took

$$X_{\alpha,\lambda}(a,t) = \zeta_{\alpha,\lambda}t, \qquad x = \zeta t \tag{51}$$

and used (39). The l.h.s. of (50) can be directly computed using that $g_{\alpha,\lambda}(\zeta)$ fulfils the continuity equation (30). In particular we find

$$\int_{\zeta_{\alpha,\lambda}}^{\Phi_{\alpha,\lambda}(\zeta)} \mathrm{d}y \; g_{\alpha,\lambda}(y) = (\Phi_{\alpha,\lambda}(\zeta) - v_{\alpha,\lambda}(\Phi_{\alpha,\lambda}(\zeta)))g_{\alpha,\lambda}(\Phi_{\alpha,\lambda}(\zeta)). \tag{52}$$

Plugging this into (50) gives

$$g_{\alpha,\lambda}(\Phi_{\alpha,\lambda}(\zeta)) = \frac{\zeta}{\Phi_{\alpha,\lambda}(\zeta) - v_{\alpha,\lambda}(\Phi_{\alpha,\lambda}(\zeta))}(\theta_H(\zeta)g_{\alpha,\lambda}^{(R)} + \theta_H(-\zeta)g_{\alpha,\lambda}^{(L)}). \tag{53}$$

All this has a very simple physical interpretation. This equation states that, since (48) is conserved, $g_{\alpha,\lambda}(\Phi_{\alpha,\lambda}(\zeta))$ is renormalised by the factor

$$\frac{\zeta}{\Phi_{\alpha,\lambda}(\zeta) - v_{\alpha,\lambda}(\Phi_{\alpha,\lambda}(\zeta))} \tag{54}$$

to account for changes in the "volume" $X_{\alpha,\lambda}(x,t) - X_{\alpha,\lambda}(a,t)$. For instance, if the trajectories are straight lines the volume is constant and, accordingly, we have

$$\frac{\zeta}{\Phi_{\alpha,\lambda}(\zeta) - v_{\alpha,\lambda}(\Phi_{\alpha,\lambda}(\zeta))} = 1. \tag{55}$$

Finally, note that (53) can be equivalently expressed as

$$g_{\alpha,\lambda}(\zeta) = \frac{Z_{\alpha,\lambda}(\zeta)}{\zeta - v_{\alpha,\lambda}(\zeta)}(\theta_H(Z_{\alpha,\lambda}(\zeta))g_{\alpha,\lambda}^{(R)} + \theta_H(-Z_{\alpha,\lambda}(\zeta))g_{\alpha,\lambda}^{(L)}). \tag{56}$$

## 5 Prediction for the entropy dynamics

In this section we derive the quasiparticle picture's prediction for the entanglement dynamics after a quench from a piecewise homogeneous initial condition (21). Specifically, we assume that $\hat{\rho}_{L(R)}$ are both pure, low-entangled, states producing pairs of entangled quasiparticles. Namely, we assume that we can use (1).

As noted in the previous section, for large enough times after a bipartite quench, we can make the following replacement (*cf.* the discussion after (41))

$$X_{\alpha,\lambda}(x,t) \to t\, \Phi_{\alpha,\lambda}(\zeta), \tag{57}$$

where the function $\Phi_{\alpha,\lambda}(\zeta)$ is defined in (40). Plugging this expression into (1) and changing variables to the ray $\zeta = x/t$ we have

$$S_A(t) = t \sum_\alpha \int d\lambda \int d\zeta\, f_{\alpha,\lambda}(\zeta t)\chi_{[\zeta_1,\zeta_2]}(\Phi_{\alpha,\lambda}(\zeta))(1 - \chi_{[\zeta_1,\zeta_2]}(\Phi_{\alpha,-\lambda}(\zeta))), \tag{58}$$

where we introduced $\zeta_{1,2} \equiv x_{1,2}/t$. The identification (57), however, is not sufficient. To make Eq. (58) predictive we also have to determine the weight function $f_{\alpha,\lambda}(\zeta)$, i.e., the contribution of the pair of quasiparticles $\{(\alpha,\lambda),(\alpha,-\lambda)\}$ to the entanglement entropy.

Here we conjecture that the entanglement entropy is given by the Yang-Yang entropy $S_{\alpha,\lambda}^{YY}(\pm\infty)$ of the quasiparticles created initially in the two leads. This means

$$f_{\alpha,\lambda}(\zeta) = S_{\alpha,\lambda}^{YY}(-\infty)\theta_H(-\zeta) + S_{\alpha,\lambda}^{YY}(\infty)\theta_H(\zeta). \tag{59}$$

As discussed in Sec. 3.1, $S_{\alpha,\lambda}^{YY}(\pm\infty)$ is the density of Yang-Yang entropy of the stationary states reached after the quench from the homogeneous initial conditions on left and right leads. The ansatz (59) is motivated by two main observations. First, Eq. (59) has been verified for quenches from piecewise homogeneous initial states in free systems [65]. Second, for homogeneous quenches Eq. (59) holds true even in the presence of interactions [33].

Equation (58), supplemented with (59), gives a complete quasiparticle prediction for the dynamics of the entanglement entropy in the interacting case. It is, however, very complicated to evaluate, as it involves the determination of both $v_{\alpha,\lambda}(\zeta)$ and $\Phi_{\alpha,\lambda}(\zeta)$ for all values of $\zeta$. In the following we show that it is possible to drastically simplify Eq. (58). We begin by changing variables from $\zeta$ to $\phi = \Phi_{\alpha,\lambda}(\zeta)$

$$S_A(t) = t \sum_\alpha \int d\lambda \int \frac{d\phi}{\Phi'_{\alpha,\lambda}(Z_{\alpha,\lambda}(\phi))} f_{\alpha,\lambda}(Z_{\alpha,\lambda}(\phi))\chi_{[\zeta_1,\zeta_2]}(\phi)(1 - \chi_{[\zeta_1,\zeta_2]}(\Phi_{\alpha,-\lambda}(Z_{\alpha,\lambda}(\phi))), \tag{60}$$

where the function $Z_{\alpha,\lambda}(\phi)$ is defined in (41) and $\Phi'_{\alpha,\lambda}(\zeta) = \partial_\zeta \Phi_{\alpha,\lambda}(\zeta)$. Then we note

$$\frac{f_{\alpha,\lambda}(Z_{\alpha,\lambda}(\phi))}{\Phi'_{\alpha,\lambda}(Z_{\alpha,\lambda}(\phi))} = \frac{Z_{\alpha,\lambda}(\phi)}{\phi - v_{\alpha,\lambda}(\phi)} f_{\alpha,\lambda}(Z_{\alpha,\lambda}(\phi)) = S_{\alpha,\lambda}^{YY}(\zeta), \tag{61}$$

where in the first step we used (43) and in the second (53). Finally, we use the identity (see Appendix B for the proof)

$$\theta_H(\Phi_{\alpha,-\lambda}(Z_{\alpha,\lambda}(\phi)) - \zeta_2) = \theta_H(\phi - J_{\alpha,\lambda}(\zeta_2)). \tag{62}$$

In (62) we introduced the function $J_{\alpha,\lambda}(\zeta)$, which characterises the trajectory of entangled quasiparticles pairs. Specifically, $J_{\alpha,\lambda}(\zeta)t$ gives the position at time $t$ of the quasiparticle $(\alpha,\lambda)$ starting at the same point as the particle $(\alpha,-\lambda)$ that at time $t$ is at position $\zeta t$ (see Fig. 3 for a pictorial representation). In formulae we have

$$J_{\alpha,\lambda}(\zeta) = \theta_H(\zeta_{\alpha,-\lambda} - \zeta)\Phi_{\alpha,\lambda}\left( [v^{\min} - v_{\alpha,-\lambda}(-\infty)] \exp\left[ \int_{v^{\min}}^{\zeta} \frac{dz}{z - v_{\alpha,-\lambda}(z)} \right] \right)$$
$$+ \theta_H(\zeta - \zeta_{\alpha,-\lambda})\Phi_{\alpha,\lambda}\left( [v^{\max} - v_{\alpha,-\lambda}(\infty)] \exp\left[ \int_{\zeta}^{v^{\max}} \frac{dz}{v_{\alpha,-\lambda}(z) - z} \right] \right). \tag{63}$$

Note that one trivially has

$$J_{\alpha,\lambda}(J_{\alpha,-\lambda}(\zeta)) = \zeta. \tag{64}$$

Putting all together we find the following expression for the entanglement entropy

$$S_A(t) = t \sum_\alpha \int d\lambda \left[ \theta_H(\zeta_2 - J_{\alpha,\lambda}(\zeta_2)) \int_{\max(\zeta_1, J_{\alpha,\lambda}(\zeta_2))}^{\zeta_2} d\phi \; S_{\alpha,\lambda}^{YY}(\phi) + \theta_H(J_{\alpha,\lambda}(\zeta_1) - \zeta_1) \int_{\zeta_1}^{\min(\zeta_2, J_{\alpha,\lambda}(\zeta_1))} d\phi \; S_{\alpha,\lambda}^{YY}(\phi) \right]. \tag{65}$$

This expression can be simplified further by using the continuity equation (30) for the Yang-Yang entropy density. This allows one to perform the integral over $\phi$ explicitly

$$S_A(t) = t \sum_\alpha \int d\lambda \left\{ \theta_H(\zeta_2 - J_{\alpha,\lambda}(\zeta_2))(\phi - v_{\alpha,\lambda}(\phi)) S_{\alpha,\lambda}^{YY}(\phi) \Big|_{\phi=\max(\zeta_1, J_{\alpha,\lambda}(\zeta_2))}^{\phi=\zeta_2} \right.$$
$$\left. + \theta_H(J_{\alpha,\lambda}(\zeta_1) - \zeta_1)(\phi - v_{\alpha,\lambda}(\phi)) S_{\alpha,\lambda}^{YY}(\phi) \Big|_{\phi=\zeta_1}^{\phi=\min(\zeta_2, J_{\alpha,\lambda}(\zeta_1))} \right\}. \tag{66}$$

The last step is achieved by means of the following identity (see Appendix B for the proof)

$$(\zeta - v_{\alpha,\lambda}(\zeta)) S_{\alpha,\lambda}^{YY}(\zeta) = (J_{\alpha,-\lambda}(\zeta) - v_{\alpha,-\lambda}(J_{\alpha,-\lambda}(\zeta))) S_{\alpha,-\lambda}^{YY}(J_{\alpha,-\lambda}(\zeta)). \tag{67}$$

Using (67) we can finally rewrite (66) as follows

$$S_A(t) = t \sum_\alpha \int d\lambda \left\{ \text{sgn}(J_{\alpha,-\lambda}(\zeta_1) - \zeta_1)\text{sgn}(\zeta_2 - J_{\alpha,-\lambda}(\zeta_1))(\zeta_1 - v_{\alpha,\lambda}(\zeta_1)) S_{\alpha,\lambda}^{YY}(\zeta_1) \right.$$
$$\left. - \text{sgn}(J_{\alpha,-\lambda}(\zeta_2) - \zeta_1)\text{sgn}(\zeta_2 - J_{\alpha,-\lambda}(\zeta_2))(\zeta_2 - v_{\alpha,\lambda}(\zeta_2)) S_{\alpha,\lambda}^{YY}(\zeta_2) \right\}, \tag{68}$$

where we assumed $\zeta_1 < \zeta_2$. Contrary to (58), (68) does not require to integrate over $\zeta$; Eq. (68) is one of the main results of this paper.

The terms on the two lines of (68) correspond to processes where the quasiparticle $(\alpha, \lambda)$ crosses the boundary at $\zeta_1$ and $\zeta_2$ respectively. Let us look more closely at one of them, say the one on the first line. First we note that the contribution is positive only if the final configuration has only one of the two quasiparticles in the system, namely if the particle with rapidity $\lambda$ enters the system and its companion is outside or if the particle of rapidity $\lambda$ exits the system and its companion is inside. The contributions with final configurations featuring both quasiparticles inside or outside are instead weighted with a negative sign. This is in perfect agreement with our intuition: the entanglement increases when a pair is shared. Second, we note that the numerical value of the contribution is (*cf.* Eq. (61))

$$|x_1 - t v_{\alpha,\lambda}(\zeta_1)| S_{\alpha,\lambda}^{YY}(\zeta_1) = t |Z_{\alpha,\lambda}(\zeta_1)| f_{\alpha,\lambda}(Z_{\alpha,\lambda}(\zeta_1)). \tag{69}$$

This is nothing but the number of pairs $(\alpha, \lambda), (\alpha, -\lambda)$ for which $(\alpha, \lambda)$ crossed the border at $x_1$ before time $t$ times $f_{\alpha,\lambda}(Z_{\alpha,\lambda}(\zeta_1))$, the contribution of a single pair. This is, again, in agreement with our expectations. Analogous considerations hold for the term on the second line.

In Section 6 we discuss further qualitative features of the result (68), focusing on the dynamics of the entanglement of the semi-infinite chain. Before specialising (68) to that situation, however, we provide some consistency checks of its validity.

## 5.1 Check I: Infinite time limit on a fixed ray

As a first check of (68), we consider the evolution of the entropy when the size of the subsystem $A$ does not scale with $t$; in this case the entire subsystem is described by a single ray, say $\zeta_1$. At infinite time the entanglement entropy is expected to coincide with the entropy of the LQSS that describes the steady state at ray $\zeta_1$, in complete analogy with what happens after homogeneous quenches [33, 34, 40–43]. Let us then verify that (68) is consistent with this picture. Fixing $A = [\zeta_1 t, \zeta_2 t]$, with

$$\zeta_2 = \zeta_1 + \frac{\ell}{t} \tag{70}$$

in Eq. (68) and taking the infinite time limit with fixed subsystem size $\ell$, we find

$$\lim_{t\to\infty} \frac{S_A(t)}{\ell} = \lim_{t\to\infty} \frac{t}{\ell} \sum_\alpha \int d\lambda \left\{ \frac{\ell}{t} \left( S_{\alpha,\lambda}^{YY}(\zeta_1) + \zeta_1 \partial_{\zeta_1} S_{\alpha,\lambda}^{YY}(\zeta_1) - \partial_{\zeta_1} \left( v_{\alpha,\lambda}(\zeta_1) S_{\alpha,\lambda}^{YY}(\zeta_1) \right) \right) \right\}$$

$$= \sum_\alpha \int d\lambda \, S_{\alpha,\lambda}^{YY}(\zeta_1), \tag{71}$$

which is the expected result. In the second step we used that $S_{\alpha,\lambda}^{YY}(\zeta)$ fulfils the continuity equation (20).

## 5.2 Check II: A lead with sub-extensive entropy

Another interesting case is when one of the homogeneous states joined in the bipartite quench protocol, say that on the right, has sub-extensive entropy ($S_{\alpha,\lambda}^{YY}(\infty) = 0$). This setup has been investigated in Ref. [61] for quenches in the XXZ chain.

   In this case we expect the entanglement entropy to be independent of the position of the system's right boundary, as long as it cannot be reached by quasiparticles coming from the left lead. In the following we show that this expectation is confirmed by Eq. (68). We first note that, as a special case of (17), the Yang-Yang entropy density satisfies

$$\phi \partial_\phi \left[ \frac{S_{\alpha,\lambda}^{YY}(\phi)}{\rho_{\alpha,\lambda}^t(\phi)} \right] = v_{\alpha,\lambda}(\phi) \partial_\phi \left[ \frac{S_{\alpha,\lambda}^{YY}(\phi)}{\rho_{\alpha,\lambda}^t(\phi)} \right], \tag{72}$$

where $\rho_{\alpha,\lambda}^t(\phi)$ is the total root density defined in (9). In the specific case considered we have $S_{\alpha,\lambda}^{YY}(\infty) = 0$, so we find

$$S_{\alpha,\lambda}^{YY}(\phi) = \rho_{\alpha,\lambda}^t(\phi) \theta_H(\zeta_{\alpha,\lambda} - \phi) \frac{S_{\alpha,\lambda}^{YY}(-\infty)}{\rho_{\alpha,\lambda}^t(-\infty)}. \tag{73}$$

Considering an interval $A = [\zeta_1 t, \zeta_2 t]$ with $\zeta_2 > v^{\max}$ (*cf.* Point (i) in Sec. 4), and plugging (73) into the semiclassical expression (68), we find

$$S_A = t \sum_\alpha \int d\lambda \left\{ \mathrm{sgn}(\zeta_1 - J_{\alpha,\lambda}(\zeta_1)) \mathrm{sgn}(J_{\alpha,\lambda}(\zeta_2) - \zeta_1) \right.$$

$$\left. \times (\zeta_1 - v_{\alpha,\lambda}(\zeta_1)) \theta_H(\zeta_{\alpha,\lambda} - \zeta_1) S_{\alpha,\lambda}^{YY}(\zeta_1) \right\}. \tag{74}$$

Here we used $\mathrm{sgn}(J_{\alpha,-\lambda}(\zeta) - \zeta) = \mathrm{sgn}(\zeta - J_{\alpha,\lambda}(\zeta))$, which is a consequence of the monotonicity of the functions $J_{\alpha,\lambda}(\zeta)$ and of (64). We now show that the expression in (74) does not depend on $\zeta_2$ as long as $\zeta_2 > v^{\max}$. First, from the definition (63) of $J_{\alpha,\lambda}(\zeta)$ it follows

$$J_{\alpha,\lambda}(\zeta_2) = \Phi_{\alpha,\lambda}(\zeta_2 - v_{\alpha,-\lambda}(\infty)), \qquad \zeta_2 > v^{\max}, \tag{75}$$

where we used that for $\zeta_2 > v^{\mathrm{max}}$ the quasiparticles velocities do not depend on $\zeta$. Second, we observe

$$
\begin{aligned}
\mathrm{sgn}(\Phi_{\alpha,\lambda}(\zeta_2 - v_{\alpha,-\lambda}(\infty)) - \zeta_{\alpha,\lambda}) &= \mathrm{sgn}(\zeta_2 - v_{\alpha,-\lambda}(\infty) - Z_{\alpha,\lambda}(\zeta_{\alpha,\lambda})) \\
&= \mathrm{sgn}(\zeta_2 - v_{\alpha,-\lambda}(\infty)) = 1.
\end{aligned}
\tag{76}
$$

Here we used that both $Z_{\alpha,\lambda}$ and $\Phi_{\alpha,\lambda}$ are monotonic functions of their arguments and can then be applied to both members of a difference in the argument of the sgn function. We also used that $\Phi_{\alpha,\lambda} = Z_{\alpha,\lambda}^{-1}$, and $Z_{\alpha,\lambda}(\zeta_{\alpha,\lambda}) = 0$ (see Section 4). Putting all together we find $J_{\alpha,\lambda}(\zeta_2) > \zeta_{\alpha,\lambda}$. Combining this with the step function $\theta_H(\zeta_{\alpha,\lambda} - \zeta_1)$ appearing in the integrand of (74), we then conclude

$$
\mathrm{sgn}(J_{\alpha,\lambda}(\zeta_2) - \zeta_1)\theta_H(\zeta_{\alpha,\lambda} - \zeta_1) = \theta_H(\zeta_{\alpha,\lambda} - \zeta_1),
\tag{77}
$$

which implies that (74) does not depend on $\zeta_2$.

# 6 Entanglement entropy of a semi-infinite interval: Entanglement production rate

Here we focus on the entanglement entropy of a semi-infinite interval. There are several reasons why this quantity is interesting. First, it is much easier to calculate via direct numerical methods, such as tDMRG (*cf.* Sec. 7). Moreover, in contrast with Formula (68), it is much simpler to evaluate. Specifically, (68) requires determining the functions $J_{\alpha,\lambda}(\zeta)$ (see Sec. 4), which relate the trajectories of the quasiparticles forming an entangled pair. As we now show, this is not the case for the entanglement between two semi-infinite chains.

The starting point to derive the semiclassical prediction for the entanglement production rate is obtained from Eq. (68) by considering the limit $\zeta_2 \to \infty$, and neglecting the entropy contribution associated with the boundary at $\zeta_2 t$. The result reads as

$$
S_{[\zeta t,\infty]}(t) = t \sum_\alpha \int \mathrm{d}\lambda \, \mathrm{sgn}(J_{\alpha,-\lambda}(\zeta) - \zeta)(\zeta - v_{\alpha,\lambda}(\zeta))S_{\alpha,\lambda}^{YY}(\zeta).
\tag{78}
$$

Remarkably, under some general assumptions on the velocity field $v_{\alpha,\lambda}(\zeta)$, this formula can be further simplified, completely removing the dependence on $J_{\alpha,\lambda}(\zeta)$. To that aim, we use the following properties of the group velocities

1. $v_{\alpha,\lambda}(\pm\infty)$ are differentiable, periodic functions of $\lambda$ with period $\Lambda$;

2. $v_{\alpha,\lambda}(\pm\infty)$ are odd functions of $\lambda$;

3. $v_{\alpha,\lambda}(\pm\infty)$ have a single maximum in $[-\Lambda/2, \Lambda/2]$.

It is possible to show that 1.–3. imply that the trajectories of the entangled quasiparticles $(\alpha, \lambda)$ and $(\alpha, -\lambda)$ originating in the same point do not cross during the dynamics (see Appendix C). This implies

$$
\mathrm{sgn}(J_{\alpha,-\lambda}(\zeta) - \zeta) = \mathrm{sgn}(v_{\alpha,-\lambda}(\sigma_{\alpha,\lambda}(\zeta)\infty) - v_{\alpha,\lambda}(\sigma_{\alpha,\lambda}(\zeta)\infty)) = -\mathrm{sgn}(v_{\alpha,\lambda}(\sigma_{\alpha,\lambda}(\zeta)\infty)),
\tag{79}
$$

where we defined

$$
\sigma_{\alpha,\lambda}(\zeta) \equiv \mathrm{sgn}(Z_{\alpha,\lambda}(\zeta)) = \mathrm{sgn}(\zeta - \zeta_{\alpha,\lambda}) = \mathrm{sgn}(\zeta - v_{\alpha,\lambda}(\zeta)).
\tag{80}
$$

In the first step of (79) we used that, since the particles never cross, the sign of $J_{\alpha,-\lambda}(\zeta)-\zeta$ is the sign of the difference of the initial velocities ($\sigma_{\alpha,\lambda}(\zeta)$ discriminates whether these velocities are computed in the left or in the right state). In the second step we used that the velocities in the left and right macrostates are odd functions of $\lambda$. Finally, the last equality in (80) is due to

$$\zeta > \zeta_{\alpha,\lambda} \quad \Leftrightarrow \quad \zeta > v_{\alpha,\lambda}(\zeta), \tag{81}$$

which is a consequence of Assumption 1 and of the continuity in $\zeta$ of $v_{\alpha,\lambda}(\zeta)$.

We remark that the conditions 1.–3. are not very restrictive. For example, they hold for all the quenches considered so far in the XXZ chain. In particular, they are verified in all the bipartite quenches in the XXZ chain considered in this work.

After plugging (79) in (78) we obtain that the entanglement entropy grows linearly as

$$S_{[\zeta t,\infty]}(t) = t \sum_{\alpha} \int d\lambda \, \mathrm{sgn}(v_{\alpha,\lambda}(\sigma_{\alpha,\lambda}(\zeta)\infty))(v_{\alpha,\lambda}(\zeta)-\zeta)S_{\alpha,\lambda}^{YY}(\zeta). \tag{82}$$

This equation is our second main result, and it expresses the entanglement production rate $S'_{\mathrm{ent}}$, i.e., the slope of the linear growth in (82) solely in terms of quantities evaluated at the point $\zeta$: all information about the quasiparticle trajectories has disappeared from the final expression. This means that, at least under the assumptions 1.–3., all information about the entanglement-entropy spreading is encoded in the local equilibrium state at ray $\zeta$.

Equation (82) contains highly non-trivial effects of the interaction. To describe them, let us focus on the simplified case

$$\mathrm{sgn}(v_{\alpha,\lambda}(+\infty)) = \mathrm{sgn}(v_{\alpha,\lambda}(-\infty)) = \mathrm{sgn}(\lambda). \tag{83}$$

Once again, we stress that condition (83) holds for all the quenches in the XXZ chain that have been considered so far in the literature. In addition, we specialise (82) to the case $\zeta = 0$, namely we consider the entropy between the two leads in Fig. 1. In this case we find

$$\begin{aligned} S_{[0,\infty]}(t) &= t \sum_{\alpha} \int d\lambda \, \mathrm{sgn}(\lambda) v_{\alpha,\lambda}(0) S_{\alpha,\lambda}^{YY}(0) \\ &= t \sum_{\alpha} \int_{\lambda>0} d\lambda \left( v_{\alpha,\lambda}(0) S_{\alpha,\lambda}^{YY}(0) - v_{\alpha,-\lambda}(0) S_{\alpha,-\lambda}^{YY}(0) \right). \end{aligned} \tag{84}$$

In this expression we can distinguish four possible "processes"

(a) $\mathrm{sign}(v_{\alpha,\lambda}(0)) = 1, \quad \mathrm{sign}(v_{\alpha,-\lambda}(0)) = -1;$

(b) $\mathrm{sign}(v_{\alpha,\lambda}(0)) = 1, \quad \mathrm{sign}(v_{\alpha,-\lambda}(0)) = 1;$

(c) $\mathrm{sign}(v_{\alpha,\lambda}(0)) = -1, \quad \mathrm{sign}(v_{\alpha,-\lambda}(0)) = -1;$

(d) $\mathrm{sign}(v_{\alpha,\lambda}(0)) = -1, \quad \mathrm{sign}(v_{\alpha,-\lambda}(0)) = 1;$

The last case is forbidden by (83), which would imply the absurd scenario where $(\alpha,\lambda)$ is initially on the right of $(\alpha,-\lambda)$ despite the trajectories of the two particles not crossing. We are then left with the three possibilities (a)–(c).

We begin our analysis by considering case (a). This case accounts for the entangled quasiparticles that, at time $t$, are shared between the leads. Reasoning as in (69), we identify the contribution

$$t|Z_{\alpha,\lambda}(0)|f_{\alpha,\lambda}(Z_{\alpha,\lambda}(0)) + t|Z_{\alpha,-\lambda}(0)|f_{\alpha,-\lambda}(Z_{\alpha,-\lambda}(0)), \tag{85}$$

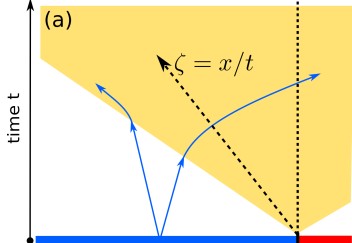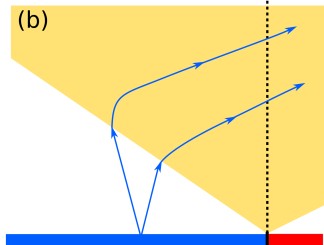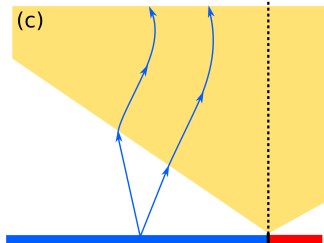

Figure 4: Possible trajectories of the entangled quasiparticles after a quench from piecewise homogenous initial state. Quasiparticle pairs are produced deep in the bulk of the two subsystems, with opposite velocities. The shaded area denotes the lightcone spreading from the interface between the two systems (vertical dotted line). Quasiparticles trajectories are linear until they hit the lightcone. Within the lightcone the quasiparticles velocities depend on the ray $\zeta \equiv x/t$. Panels (a)–(c) show the three possible trajectories of the quasiparticle pairs produced on the left of the junction. (a) Only one member of the pair reaches the interface. (b) Both members of the pair reach the interface at different times. (c) Both members are deflected back before reaching the interface.

where $\{t|Z_{\alpha,\pm\lambda}(0)|\}$ count the number of relevant pairs and $\{f_{\alpha,\lambda}(Z_{\alpha,\pm\lambda}(0))\}$ are the contributions to the entropy of each pair.

Case (b), instead, describes the situation where both quasiparticles forming an entangled pair, emitted from the left lead, eventually reach the boundary with the right lead. This implies that the velocity of one of the two entangled quasiparticles changes sign during the dynamics. Importantly, this also implies that the pairs emitted from the right do not reach the boundary and are deflected back. Indeed, two particles with the same rapidity $\lambda$ can reach $\zeta = 0$ only if they come from the same side (otherwise the velocity field in $\zeta = 0$ would not be well defined). In this case the contribution reads as

$$t|Z_{\alpha,\lambda}(0)|f_{\alpha,\lambda}(Z_{\alpha,\lambda}(0)) - t|Z_{\alpha,-\lambda}(0)|f_{\alpha,-\lambda}(Z_{\alpha,-\lambda}(0)). \tag{86}$$

Case (c) is complementary to (b): both quasiparticles emitted from the right reach the boundary and those emitted from the left are deflected back. The contribution reads as

$$-t|Z_{\alpha,\lambda}(0)|f_{\alpha,\lambda}(Z_{\alpha,\lambda}(0)) + t|Z_{\alpha,-\lambda}(0)|f_{\alpha,-\lambda}(Z_{\alpha,-\lambda}(0)). \tag{87}$$

Finally, we point out that, once crossed the junction, a particle obeying the equation of motion (31) cannot come back.

The trajectories of the entangled pairs corresponding to the three different scenario are schematically represented in Figure 4. Note that (a) is the standard case: it is realised in quenches from piecewise homogeneous initial states in free systems [65] and in quenches in interacting integrable systems evolving from homogeneous states [33, 34]. Cases (b) and (c), instead, are a landmark of the simultaneous presence of inhomogeneity and interactions. In these cases the entangled pair contributes only for a finite time, *i.e.*, only when the two quasiparticles are in different leads. After both quasiparticles crossed the boundary they do not contribute anymore to the entropy dynamics. The different scenarios (a)–(c) are illustrated in Figure 5 for some concrete bipartite quenches in the XXZ chain.

## 6.1 Entanglement versus thermodynamic entropy production rate

Let us now focus on the entanglement production rate. This quantity is defined as the slope of the linear growth of the entanglement entropy of a semi-infinite interval, or, in other words,

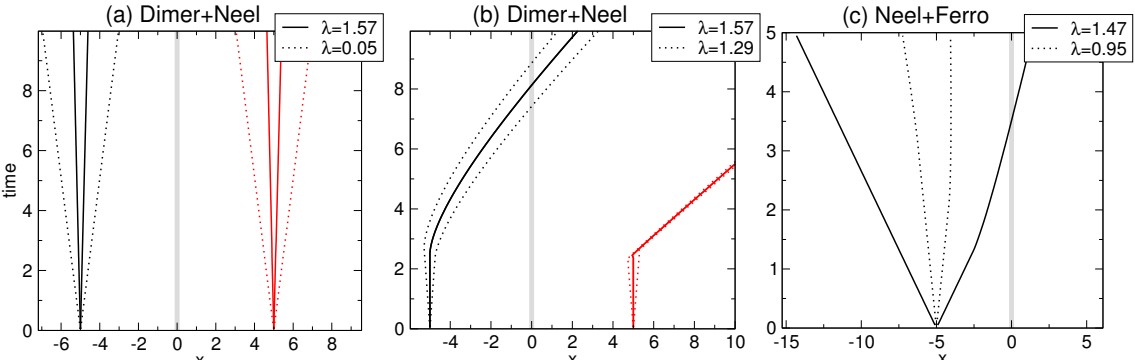

Figure 5: Trajectories of entangled quasiparticles $(\alpha, \lambda)$ and $(\alpha, -\lambda)$ after a bipartite quench in the XXZ chain. $x$ and $y$-axis show space and time, respectively. Different lines correspond to different $\lambda$s. (a) Trajectories of the quasiparticles of the first species ($\alpha = 1$) after the sudden junction of the Néel state $|N, 0\rangle$ (24) (left) and the Dimer state (23) (right). (b) Trajectories of the quasiparticles of the second species ($\alpha = 2$) after the sudden junction of the Néel state $|N, 0\rangle$ (24) (left) and the Dimer state (23) (right). Note that the trajectories of pairs corresponding to solid lines are always very close (but not coinciding). This is due to the fact that $\lambda \approx \pi/2$. (c) Trajectories of the quasiparticles of the first species ($\alpha = 1$) after the sudden junction of the Ferromagnetic state $|F, 0\rangle$ (25) (left) and the Néel state $|N, 0\rangle$ (24). The results are obtained by solving the equation of motion (31); We took $\Delta = 10$ in panels (a) and (b), and $\Delta = 5$ in panel (c).

as the time derivative of (78) at fixed ray $\zeta$. As discussed in Section 6, it reads as (*cf.* (82))

$$S'_{\text{ent},\zeta} \equiv S'_{[\zeta t,\infty]} = \sum_\alpha \int d\lambda \, \text{sgn}(v_{\alpha,\lambda}(\sigma_{\alpha,\lambda}(\zeta)\infty))(v_{\alpha,\lambda}(\zeta) - \zeta)S^{YY}_{\alpha,\lambda}(\zeta). \tag{88}$$

It is particularly instructive to compare it with the rate at which the two semi-infinite intervals exchange thermodynamic entropy. This is defined as

$$S'_{\text{th},\zeta} \equiv \sum_\alpha \int d\lambda \, |v_{\alpha,\lambda}(\zeta) - \zeta| S^{YY}_{\alpha,\lambda}(\zeta). \tag{89}$$

First of all, we observe that the exchange rate of thermodynamic entropy is an upper bound for the entanglement production rate, i.e.,

$$S'_{\text{ent},\zeta} \leq S'_{\text{th},\zeta}, \qquad \forall \zeta. \tag{90}$$

This is a direct consequence of

$$\text{sgn}(v_{\alpha,\lambda}(\sigma_{\alpha,\lambda}(\zeta)\infty))(v_{\alpha,\lambda}(\zeta) - \zeta) \leq |v_{\alpha,\lambda}(\zeta) - \zeta|. \tag{91}$$

At the specific ray $\zeta = 0$, after quenches from homogeneous initial states [33, 34] and after bipartite quenches in free models [65], the bound is saturated, $S'_{\text{ent},0} = S'_{\text{th},0}$. Indeed, in these cases the group velocity does not depend on $\zeta$, implying

$$\text{sgn}(v_{\alpha,\lambda}(\sigma_{\alpha,\lambda}(0)\infty))v_{\alpha,\lambda}(0) = \text{sgn}(v_{\alpha,\lambda}(0))v_{\alpha,\lambda}(0) = |v_{\alpha,\lambda}(0)|. \tag{92}$$

After bipartite quenches in interacting integrable systems, instead, $S'_{\text{ent},0}$ and $S'_{\text{th},0}$ are generically different. Using the simplifying assumption (83) we see that only scenario (a) in Figure 4

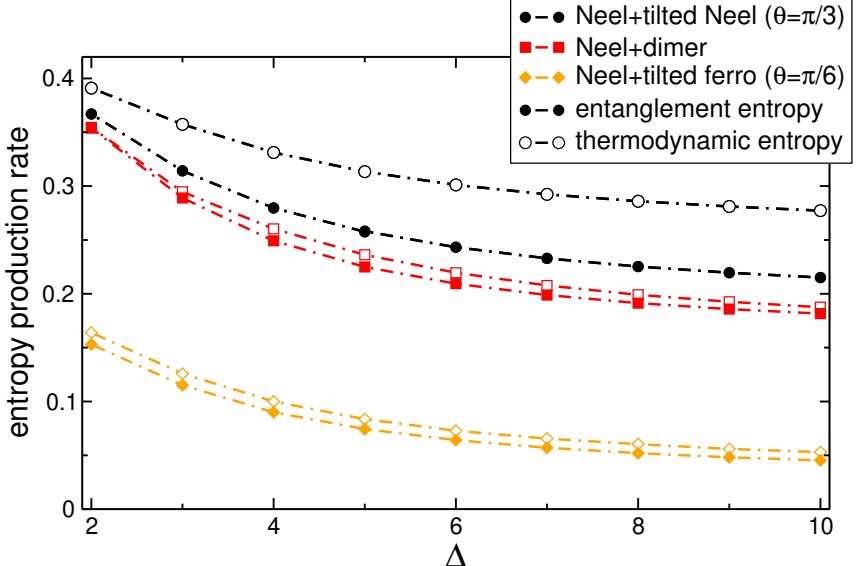

Figure 6: Entanglement (full simbols) versus thermodynamic (empty symbols) entropy production rate after the quench from a piecewise homogeneous initial state in the XXZ chain. The $x$-axis shows that chain anisotropy. Different symbols are used for different initial states obtained by joining the tilted Néel state (24), the dimer state (23), and the tilted ferromagnetic state (25). Full symbols are obtained using Eq. (88) while empty symbols are obtained using Eq. (89).

ensures $S'_{\text{ent},0} = S'_{\text{th},0}$. On the contrary, scenarios (b) and (c) imply $S'_{\text{ent},0} \neq S'_{\text{th},0}$. An illustration of the generic behaviour is given in Figure 6, which reports results for three bipartite quenches in the XXZ chain.

We point out that, if one of the scenarios (b) and (c) is allowed, the equality could hold only in the special case when the pairs reaching the junction originate on a lead with sub-extensive entropy (see Section 5.2). This is the case, for instance, after bipartite quenches in the XXZ chain if one of the two leads is prepared in the ferromagnetic state [66].

## 7 tDMRG benchmark

Here we present numerical checks of the results of Sections 5 and 6, focussing on the evolution of the half-chain entanglement entropy after a bipartite quench in the XXZ spin-1/2 chain with $\Delta > 1$ (*cf.* (4)).

We first focus on the quench from the state $|\text{N}, 0\rangle \otimes |\text{F}, \theta\rangle$ (*cf.* (24) and (25)). The results are presented in Figure 7. The different continuous curves are tDMRG results for different values of the tilting angle $\theta$ and of the chain anisotropy $\Delta$. In the simulations we considered maximal bond dimensions $\chi_{\text{max}} \approx 400$, which allowed us to reach half-chain entropy $S \approx 3$ and times $t \approx 25$. The dashed-dotted lines in the figure are the prediction (82). The agreement between the numerics and the analytical result is satisfactory for all quenches.

Let us now consider the difference between the entanglement production rate (88) and the production rate (89) of thermodynamic entropy. As demonstrated in Figure 6, this difference is generically quite small. This makes its numerical observation a highly non-trivial task. In practice, we verified that for all the quenches discussed in Fig. 7, the difference $S_{\text{ent}} - S_{\text{th}}$ is not visible within the times and system sizes accessible with tDMRG.

To accentuate the discrepancy between $S_{\text{ent}}$ and $S_{\text{th}}$, we consider the case in Fig. 6 that

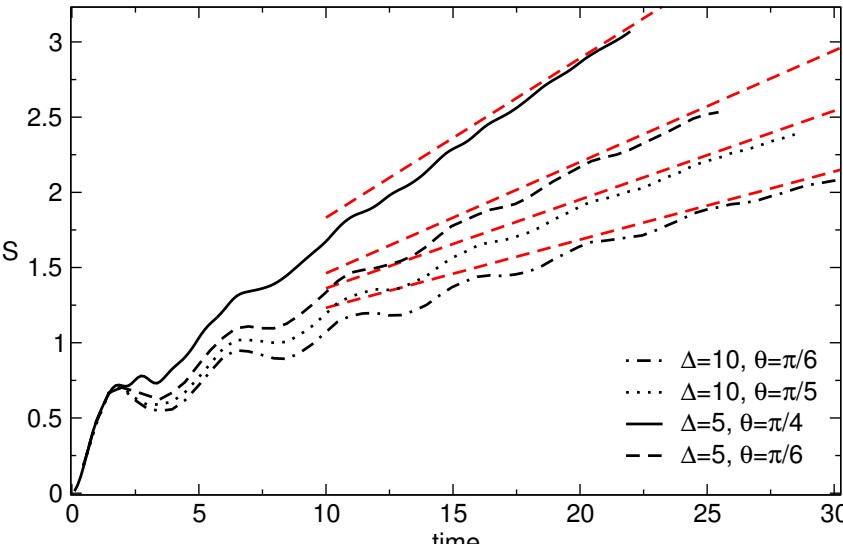

Figure 7: Entanglement dynamics after a bipartite quench in the XXZ chain. In the initial state two semi-infinite chains are prepared in the Néel state $|N, 0\rangle$ (24) (left) and the tilted ferromagnetic state (25) (right), respectively. The figure shows the half-chain entropy plotted versus the time after the quench. The different curves are tDMRG data for the chain with $\Delta = 5, 10$ and several values of $\theta$ (tilting angle). The dashed lines are linear fits. The slope of the lines is fixed by the prediction of the quasiparticle picture (82).

shows the largest difference $|S_{\text{ent}} - S_{\text{th}}|$, namely, the quench from the state $|N\rangle \otimes |N, \theta\rangle$ (*cf.* (24)). In this case, however, the entanglement growth is much faster, posing a severe limitation to the timescales accessible by tDMRG. The time evolution of the entanglement entropy after the quench for different values of the tilting angle $\theta$ and of the anisotropy $\Delta$ is reported in Figure 8. For short times the tDMRG data exhibit large finite-time effects and are not described by (82). On the other hand, for $t \gtrsim 6$ the numerical data become compatible with the slope $S'_{\text{ent}}$. Still, much larger timescales are needed to provide a robust verification of (82).

## 8 Conclusions

We investigated the dynamics of the entanglement entropy after quenches from a piecewise homogeneous initial states in interacting integrable systems. By combining the quasiparticle picture for the entanglement spreading with the GHD approach, we derived an analytic prediction for the entropy evolution after the quench. Remarkably, the entanglement production rate, *i.e.,* the growth rate of the entanglement between two half-infinite chains is described by a simple formula that we provided. This depends only on the thermodynamic macrostate (GGE) that describes local properties near the interface between the two chains at infinite time, as it was pointed out in Ref. [61]. We showed, however, that the entanglement production rate is different from the rate of exchange of thermodynamic entropy between the two half-infinite chains. This is in contrast with quenches in free-fermion models [65] and in homogeneous systems [33, 34] and it is a genuine effect of the combination of inhomogeneity and interactions.

Our work calls attention to several interesting directions for future research. An immediate one is to provide a more robust independent numerical check, going beyond the tDMRG time scales that we accessed in this work. Moreover, our analytic formula for the entanglement

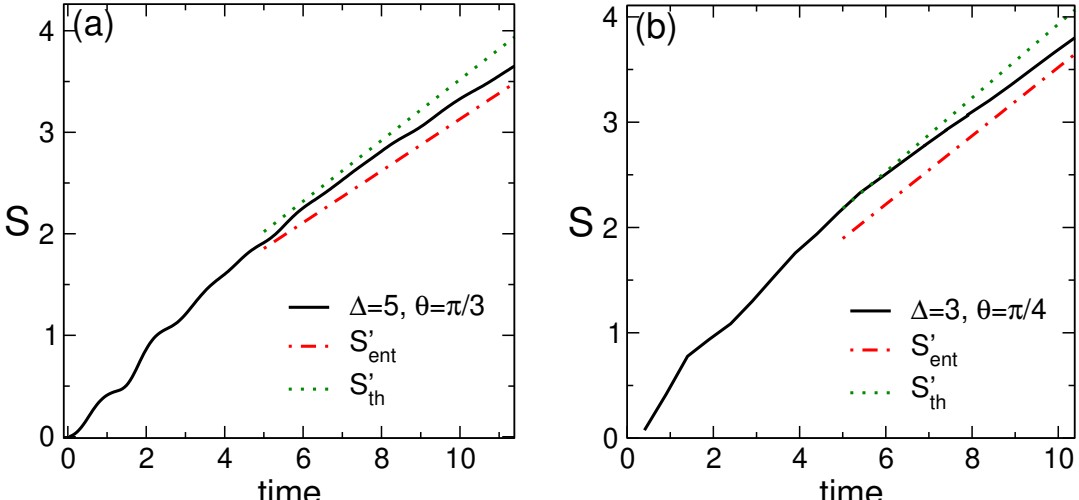

Figure 8: Entanglement dynamics after a bipartite quench in the XXZ chain. The initial state is obtained by joining the Néel state $|N, 0\rangle$ and the tilted Néel state $|N, \theta\rangle$ (24). The two panels report the dynamics for different values of the chain anisotropy $\Delta$ and the tilting angle $\theta$. The continuous lines are tDMRG results. The slope of the dashed-dotted line is $S'_{ent}$, the entanglement production rate at $\zeta = 0$ (*cf.* (88)). The slope of the dotted lines is $S'_{th}$, the exchange rate of thermodynamic entropy at $\zeta = 0$ (*cf.* (89)).

dynamics of a finite interval (*cf.* (68)) requires the quasiparticle trajectories, which have to be determined numerically. A promising alternative route is to apply the so-called "flea gas" approach [77]. There, the dynamics of out-of-equilibrium quantum systems is simulated by a gas of point-like particles travelling ballistically and scattering elastically.

Another interesting direction is to extend our framework to describe the dynamics of Rényi entropies. Indeed, as recently shown in Ref. [97], from the the dynamics of Rényi entropies one can extract that of the logarithmic negativity [98–107], which is a good entanglement measure for mixed states. Unfortunately, even if the steady-state value of the Rényi entropies is known [66, 108–110], computing their full dynamics remains a highly challenging task. A severe complication is that the thermodynamic macrostate describing the Rényi entropies does not coincide with that describing local operators and depends non-trivially on the Rényi index.

Finally, a local breaking of integrability around the interface between the two chains is expected to have dramatic effects on the entanglement production rate along any ray. We wonder, however, whether the ray $\zeta = 0$ is somehow exceptional, displaying a completely different qualitative behaviour. We leave this question to future investigations.

# Acknowledgements

B.B. and M.F. thank Pasquale Calabrese and Lorenzo Piroli for collaborations on closely related subjects.

**Funding Information** V.A. has been supported by the European Union's programme Horizon 2020 under the Marie Skłodowska-Curie grant No. 702612 – OEMBS. B.B. has been supported by the European Research Council under the Advanced Grant No. 694544 – OMNES, and by the Slovenian Research Agency (ARRS) under the grant P1-0402. M.F. has been supported by a grant LabEx PALM (ANR-10-LABX- 0039-PALM) and by the European Research Council

under the Starting Grant No. 805252 – LoCoMacro. Part of the work has been done at the Erwin Schrödinger Institute (ESI) in Vienna, during the workshop "Quantum Paths", and at the International Institute of Physics (IIP) in Natal, during the workshop "Transport in strongly correlated quantum systems".

# A Trajectories of semiclassical quasiparticles

## A.1 Properties of the velocity field

Here we prove that, for the velocity field defined by (27)–(29), the Points (i) and (ii) of Section 4 are fulfilled. Point (i) is almost trivial. For $\zeta > v^{\max}$ all functions $\vartheta_{\alpha,\lambda}(\zeta)$ defined in (27) become independent of $\zeta$. It then follows from (28) and (29) that also $\rho^t_{\alpha,\lambda}(\zeta)$ and $v_{\alpha,\lambda}(\zeta)$ become independent of $\zeta$. The same holds true for $\zeta < v^{\min}$. Let us now consider Point (ii): we will show that $v_{\alpha,\lambda}(\zeta)$ is always differentiable with bounded derivative. We start by noting that $\rho^t_{\alpha,\lambda}(\zeta)$ fulfils a continuity equation of the form (30), as it can be immediately seen from (13), (28), and (29). We then have

$$\rho^t_{\alpha,\lambda}(\zeta)\partial_\zeta v_{\alpha,\lambda}(\zeta) = (\zeta - v_{\alpha,\lambda}(\zeta))\partial_\zeta \rho^t_{\alpha,\lambda}(\zeta). \tag{93}$$

Here we always assume $\rho^t_{\alpha,\lambda}(\zeta) \neq 0$ for all $\alpha$, $\lambda$, and $\zeta$. Moreover, as a consequence of (i), we have

$$\partial_\zeta v_{\alpha,\lambda}(\zeta) = \partial_\zeta \rho^t_{\alpha,\lambda}(\zeta) = 0 \qquad \text{for} \qquad \zeta > v^{\max} \wedge \zeta < v^{\min}. \tag{94}$$

Thus, to show that $\partial_\zeta v_{\alpha,\lambda}(\zeta)$ is bounded we just need to show that $\partial_\zeta \rho^t_{\alpha,\lambda}(\zeta)$ is continuous. Taking the derivative of (28) we have

$$\partial_\zeta \rho^t_{\alpha,\lambda}(\zeta) = \sum_{\beta=1}^{N_s} \int d\mu\, T_{\alpha,\lambda;\beta,\mu}\, \rho^t_{\beta,\mu}(\zeta)\partial_\zeta \vartheta_{\beta,\mu}(\zeta) + \sum_{\beta=1}^{N_s} \int d\mu\, T_{\alpha,\lambda;\beta,\mu}\, \vartheta_{\beta,\mu}(\zeta)\partial_\zeta \rho^t_{\beta,\mu}(\zeta). \tag{95}$$

From this equation follows that $\partial_\zeta \rho^t_{\alpha,\lambda}(\zeta)$ is continuous in $\zeta$ if the driving term (the first term on the r.h.s.) is. Taking the derivative of (27) we find that the driving term reads as

$$\sum_{\beta=1}^{N_s} \int d\mu\, T_{\alpha,\lambda;\beta,\mu}\, \rho^t_{\beta,\mu}(\zeta)\partial_\zeta \vartheta_{\beta,\mu}(\zeta) = \sum_{\beta=1}^{N_s} \int d\mu\, T_{\alpha,\lambda;\beta,\mu}\, \rho^t_{\beta,\mu}(\zeta)[\vartheta^{(R)}_{\mu,\beta}(\zeta) - \vartheta^{(L)}_{\mu,\beta}(\zeta)]\delta(\zeta - \zeta_{\beta,\mu}). \tag{96}$$

The function $\zeta_{\beta,\mu}$ (defined in Assumption 1) is a continuous piecewise invertible function of $\mu$, so the sum on the r.h.s. can be written as a sum of continuous functions of $\zeta$ by integrating over the Dirac delta function. This implies that $\partial_\zeta \rho^t_{\alpha,\lambda}(\zeta)$ is a continuous function of $\zeta$ and concludes the proof.

## A.2 Proof of Eq. (38)

We start by considering (36) for $x < v^{\min} t_0$. In this case, using that $v_{\alpha,\lambda}(\zeta) = v_{\alpha,\lambda}(-\infty)$ for $\zeta < v^{\min}_\alpha(-\infty)$, we find

$$\left(x - v_{\alpha,\lambda}(-\infty)t_0\right) = t(v^{\min} - v_{\alpha,\lambda}(-\infty))\exp\left[\int_{v^{\min}}^{X_{\alpha,\lambda}(x,t)/t} \frac{d\zeta}{\zeta - v_{\alpha,\lambda}(\zeta)}\right]. \tag{97}$$

Analogously, considering $x > v^{\max} t_0$, we have

$$\left(x - v_{\alpha,\lambda}(\infty)t_0\right) = t(v^{\max} - v_{\alpha,\lambda}(\infty))\exp\left[\int_{X_{\alpha,\lambda}(x,t)/t}^{v^{\max}} \frac{d\zeta}{v_{\alpha,\lambda}(\zeta) - \zeta}\right]. \tag{98}$$

Putting all together we find

$$\frac{x}{t} = \theta(v^{\min}t_0 - x)\frac{x}{t} + \theta(v^{\max}t_0 - x)\theta(x - v^{\min}t_0)\frac{x}{t} + \theta(x - v^{\max}t_0)\frac{x}{t}$$

$$= \theta(v^{\min}t_0 - x)(v^{\min} - v_{\alpha,\lambda}(-\infty)) \exp\left[\int_{v^{\min}}^{X_{\alpha,\lambda}(x,t)/t} \frac{d\zeta}{\zeta - v_{\alpha,\lambda}(\zeta)}\right]$$

$$+ \theta(x - v^{\max}t_0)(v^{\max} - v_{\alpha,\lambda}(\infty)) \exp\left[\int_{X_{\alpha,\lambda}(x,t)/t}^{v^{\max}} \frac{d\zeta}{v_{\alpha,\lambda}(\zeta) - \zeta}\right] + O\left(\frac{t_0}{t}\right), \qquad (99)$$

where we used that, if $v^{\min}t_0 \leq x \leq v^{\max}t_0$, then $x/t = O(t_0/t)$. Using now (34) we have

$$\frac{x}{t} = \theta_H\left(\zeta_{\alpha,\lambda} - \frac{X_{\alpha,\lambda}(x,t)}{t}\right)[v^{\min} - v_{\alpha,\lambda}(-\infty)] \exp\left[\int_{v^{\min}}^{X_{\alpha,\lambda}(x,t)/t} \frac{d\zeta}{\zeta - v_{\alpha,\lambda}(\zeta)}\right]$$

$$+ \theta_H\left(\frac{X_{\alpha,\lambda}(x,t)}{t} - \zeta_{\alpha,\lambda}\right)[v^{\max} - v_{\alpha,\lambda}(\infty)] \exp\left[\int_{X_{\alpha,\lambda}(x,t)/t}^{v^{\max}} \frac{d\zeta}{v_{\alpha,\lambda}(\zeta) - \zeta}\right] + O\left(\frac{t_0}{t}\right), \quad (100)$$

which is Eq. (38).

# B  Details on the calculation of the entanglement entropy

## B.1  Proof of (62)

Using the monotonicity of $Z_{\alpha,\lambda}(\phi)$ in $\phi$ we have

$$\theta_H(\Phi_{\alpha,-\lambda}(Z_{\alpha,\lambda}(\phi)) - \zeta_2) = \theta_H(Z_{\alpha,-\lambda}\left(\Phi_{\alpha,-\lambda}(Z_{\alpha,\lambda}(\phi))\right) - Z_{\alpha,-\lambda}(\zeta_2))$$

$$= \theta_H(Z_{\alpha,\lambda}(\phi) - Z_{\alpha,-\lambda}(\zeta_2)). \qquad (101)$$

Applying Eq. (41) we find

$$\theta_H(\Phi_{\alpha,-\lambda}(Z_{\alpha,\lambda}(\phi)) - \zeta_2) \qquad (102)$$

$$= \theta_H(\zeta_{\alpha,-\lambda} - \zeta_2)\theta_H\left(Z_{\alpha,\lambda}(\phi) - [v_{\alpha,-\lambda}(-\infty) - v^{\min}] \exp\left[\int_{v^{\min}}^{\zeta_2} \frac{dz}{z - v_{\alpha,-\lambda}(z)}\right]\right)$$

$$+ \theta_H(\zeta_2 - \zeta_{\alpha,-\lambda})\theta_H\left(Z_{\alpha,\lambda}(\phi) - [v^{\max} - v_{\alpha,-\lambda}(\infty)] \exp\left[\int_{\zeta_2}^{v^{\max}} \frac{dz}{v_{\alpha,-\lambda}(z) - z}\right]\right). \qquad (103)$$

Finally, using the monotonicity in $\zeta$ of $\Phi_{\alpha,\lambda}(\zeta)$ we have

$$\theta_H(\Phi_{\alpha,-\lambda}(Z_{\alpha,\lambda}(\phi)) - \zeta_2) \qquad (104)$$

$$= \theta_H(\zeta_{\alpha,-\lambda} - \zeta_2)\theta_H\left(\phi - \Phi_{\alpha,\lambda}\left([v_{\alpha,-\lambda}(-\infty) - v^{\min}] \exp\left[\int_{v^{\min}}^{\zeta_2} \frac{dz}{z - v_{\alpha,-\lambda}(z)}\right]\right)\right)$$

$$+ \theta_H(\zeta_2 - \zeta_{\alpha,-\lambda})\theta_H\left(\phi - \Phi_{\alpha,\lambda}\left([v^{\max} - v_{\alpha,-\lambda}(\infty)] \exp\left[\int_{\zeta_2}^{v^{\max}} \frac{dz}{v_{\alpha,-\lambda}(z) - z}\right]\right)\right). \qquad (105)$$

Using the definition (63) of the function $J_{\alpha,\lambda}(\zeta)$ this equation immediately gives (62).

### B.2 Proof of (67)

Applying (53) to the Yang-Yang entropy $S^{YY}_{\alpha,\lambda}(\Phi_{\alpha,\lambda}(\zeta))$ and using $S^{YY}_{\alpha,\pm\lambda}(\pm\infty) = S^{YY}_{\alpha,\lambda}(\pm\infty)$ we find

$$(\Phi_{\alpha,\lambda}(\zeta) - v_{\alpha,\lambda}(\Phi_{\alpha,\lambda}(\zeta)))S^{YY}_{\alpha,\lambda}(\Phi_{\alpha,\lambda}(\zeta)) = (\Phi_{\alpha,-\lambda}(\zeta) - v_{\alpha,-\lambda}(\Phi_{\alpha,-\lambda}(\zeta)))S^{YY}_{\alpha,-\lambda}(\Phi_{\alpha,-\lambda}(\zeta)). \quad (106)$$

Using the arbitrariness of $\zeta$ and the definition (63) of $J_{\alpha,\lambda}(\zeta)$ this equation can be written as

$$(\zeta - v_{\alpha,\lambda}(\zeta))S^{YY}_{\alpha,\lambda}(\zeta) = (J_{\alpha,-\lambda}(\zeta) - v_{\alpha,-\lambda}(J_{\alpha,-\lambda}(\zeta)))S^{YY}_{\alpha,-\lambda}(J_{\alpha,-\lambda}(\zeta)), \quad (107)$$

which is (67).

## C  Non-crossing of quasiparticles trajectories

Here we show that the trajectories of the entangled quasiparticles with opposite rapidities $\pm\lambda$ do not cross during the dynamics. Specifically, we show that

$$\Phi_{\alpha,\lambda}(\zeta) \neq \Phi_{\alpha,-\lambda}(\zeta), \quad (108)$$

where $\Phi_{\alpha,\lambda}(\zeta)$ is defined in (40). To prove (108) we use the following assumptions on the velocity field $v_{\alpha,\lambda}(\pm\infty)$

1. $v_{\alpha,\lambda}(\pm\infty)$ are differentiable, periodic functions of $\lambda$ with period $\Lambda$.

2. $v_{\alpha,\lambda}(\pm\infty)$ are odd functions of $\lambda$.

3. $v_{\alpha,\lambda}(\pm\infty)$ have a single maximum in $[-\Lambda/2, \Lambda/2]$.

The first step is to take the $\lambda$-derivative of (40) at fixed $\zeta$

$$\partial_\lambda \Phi_{\alpha,\lambda}(\zeta) = [\Phi_{\alpha,\lambda}(\zeta) - v_{\alpha,\lambda}(\Phi_{\alpha,\lambda}(\zeta))]\Big[\frac{v'_{\alpha,\lambda}(\infty)\theta_H(\zeta)}{v^{max} - v_{\alpha,\lambda}(\infty)} + \frac{v'_{\alpha,\lambda}(-\infty)\theta_H(-\zeta)}{v^{min} - v_{\alpha,\lambda}(-\infty)}\Big], \quad (109)$$

where we used

$$\theta_H(\pm\zeta) = \theta_H(\pm\zeta_{\alpha,\lambda} \mp \Phi_{\alpha,\lambda}(\zeta)). \quad (110)$$

Let us define $\bar{\lambda}_{\alpha,+}$ and $\bar{\lambda}_{\alpha,-}$ as the rapidities corresponding to the maximum of $v_{\alpha,\lambda}(\infty)$ and $v_{\alpha,\lambda}(-\infty)$ respectively. We then distinguish four cases depending on the sign of $\zeta$ and of $|\lambda| - |\bar{\lambda}_{\alpha,\pm}|$.

$$\text{(i) } \zeta > 0 \text{ and } |\lambda| < |\bar{\lambda}_{\alpha,+}|, \qquad \text{(ii) } \zeta > 0 \text{ and } |\lambda| > |\bar{\lambda}_{\alpha,+}|, \quad (111)$$

$$\text{(iii) } \zeta < 0 \text{ and } |\lambda| < |\bar{\lambda}_{\alpha,-}|, \qquad \text{(iv) } \zeta < 0 \text{ and } |\lambda| > |\bar{\lambda}_{\alpha,-}|. \quad (112)$$

We start with the proof of case (i). By integrating (109) from $-\lambda$ to $\lambda$, we find

$$\Phi_{\alpha,\lambda}(\zeta) - \Phi_{\alpha,-\lambda}(\zeta) = \int_{-\lambda}^{\lambda} d\mu [\Phi_{\alpha,\mu}(\zeta) - v_{\alpha,\mu}(\Phi_{\alpha,\mu}(\zeta))]\frac{v'_{\alpha,\mu}(\infty)}{v^{max} - v_{\alpha,\mu}(\infty)}. \quad (113)$$

The integrand has fixed sign in the interval $[-\lambda, \lambda]$. This is because we have that for any $\zeta$, $\Phi_{\alpha,\mu}(\zeta) - v_{\alpha,\mu}(\Phi_{\alpha,\mu}(\zeta)) > 0$, $v'_{\alpha,\mu}(\infty)$ has fixed sign $\forall \mu \in [-\lambda, \lambda]$, and $v^{max} - v_{\alpha,\mu}(\infty) > 0$, for

any $\mu$. Thus, we conclude that (108) holds. The case (ii) is treated similarly. By integrating (109) from $\lambda$ to $\Lambda/2$ and from $-\Lambda/2$ to $-\lambda$ we find

$$\Phi_{\alpha,\lambda}(\zeta) - \Phi_{\alpha,-\lambda}(\zeta) = \Phi_{\alpha,\lambda}(\zeta) - \Phi_{\alpha,\Lambda/2}(\zeta) + \Phi_{\alpha,-\Lambda/2}(\zeta) - \Phi_{\alpha,-\lambda}(\zeta)$$
$$= -\left( \int_{-\Lambda/2}^{-\lambda} + \int_{\lambda}^{\Lambda/2} \right) d\mu \left[ \Phi_{\alpha,\mu}(\zeta) - v_{\alpha,\mu}(\Phi_{\alpha,\mu}(\zeta)) \right] \frac{v'_{\alpha,\mu}(\infty)}{v^{\max} - v_{\alpha,\mu}(\infty)}. \quad (114)$$

Here we used $\Phi_{\alpha,-\Lambda/2}(\zeta) = \Phi_{\alpha,\Lambda/2}(\zeta)$, which follows from the definition (40) and the periodicity of $v_{\alpha,\mu}(\pm\infty)$. As for case (i), the integrand has fixed sign in the integration interval, implying that (108) holds true. Finally, the remaining cases (iii) and (iv) are treated in a completely analogous way.

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
