# Peer review of "Entanglement evolution and generalised hydrodynamics: interacting integrable systems"

_SciPost Physics, doi:SciPost Phys. 7, 005 (2019)_

## Round 2 · Referee Report · Anonymous (Referee 1) · 2019-5-1

Strengths
nicely written
general and strong result
good numerical checks
timely topic
Weaknesses
none
Report
In this paper, the authors study the evolution of entanglement entropy in inhomogeneous states of integrable quantum models. They combine the ideas of pair particle production with the formalism of generalised hydrodynamics, and obtain exact formulae for the growth of entanglement entropy in a region as function of time, valid at large times. The idea of the particle pair production makes the analysis quite clear, and it is a matter of combining this with the nontrivial curved trajectories that generically emerge in generalised hydrodynamics. They study the situation of a bipartite initial state, where the trajectories can be studied in more details. Surprising simplifications occur in the formula, although it is still quite complicated. The main formula is expressed in quite some generality within GHD, and has a clear physical interpretation in terms of quasiparticles entering or not the region of entanglement. It turns out that the entanglement entropy growth is different from the thermodynamic entropy of the growing region. This is surprising at first, but quite understandable with the fact that quasiparticles have curved trajectories. The authors check in various limits that the formula reproduces the known or expected results, and make adequate numerical verifications in the Heisenberg chain.
The paper is very well written, very readable. It overviews the various ideas and theories quite well, and the calculations are involved but clear. I have not checked all calculations, but those I have checked appear to be correct, and to account for all necessary subtleties. The only small improvement I would suggest is in the explanations around eq. 12. For appropriate context, I would suggest to mention that this is the hydrodynamic approximation. It is also important to mention that the result holds for x scaling with t as well, as such a scaling is what is used throughout the paper (and maybe mention that on the right-hand side, the only x,t dependence is in the density matrix; the x in the operator is not relevant). Besides, I don’t have any particular comment; I think the paper is perfect for publication as it is.
Requested changes
explanations around equation 12, see report.

---

## Round 2 · Referee Report · Anonymous (Referee 2) · 2019-5-27

Strengths
- timely topic
- interesting physical mechanism and differences from noninteracting case
- well written
Weaknesses
- somewhat too technical at points
Report
The authors study entanglement evolution in 1D integrable systems in
the framework of the recently developed generalized hydrodynamics.
The main goal of the paper is to extend previous results on the
entanglement entropy after a global quench to inhomogeneous situations,
with the same physical picture of quasiparticle pairs carrying entanglement.
In particular, the focus is on initial states with a bipartite tensor
product structure, where both left/right hand sides correspond to
homogeneous states. In such a situation previous results have been
derived for the noninteracting case by some of the authors.
The main difficulty of the interacting problem is that the quasiparticles
have now a nonlinear motion due to the nontrivial background created by
other quasiparticles. In fact, the trajectories become nonlinear once
a propagating quasiparticle enters the light-cone region expanding from
the junction of the two half-chains. While the general formula (68) for
a segment requires to keep track of the particle-pair motion, for the
semi-infinite case it is shown that, under some mild assumptions on
the dressed velocities, information on the trajectories is not required.
The main difference w.r.t. the noninteracting case is, however, that
the velocities of a particle pair created far away from the junction
might change completely whilst crossing the inhomogeneous zone.
In fact, both of them could reach the junction or could be completely
deflected. These processes then do not contribute to the entanglement
generation and it is shown that the thermodynamic entropy growth rate
is always an upper bound of the former. The predictions are then tested
for inhomogeneous quenches of the XXZ chain, with a reasonably good
agreement obtained from the DMRG simulations of the entropy growth.
I believe that this is a very nice result and the paper is well written,
although rather technical at some points. I found some of the derivations
hard to follow, but the final results have nevertheless a very clean
physical interpretation. The manuscript clearly deserves to be published
and I have only very few questions/comments to add.
Requested changes
(1)
In Fig. 3(c) the label should be Jα,−λt (sign missing).
(2)
In Fig. 5(b) for the solid line there is only one trajectory to be seen.
Do both particles indeed follow exactly the same trajectory?
If yes is there a simple explanation why this happens?
(3)
What would change if the group velocity did not have only a single maximum?

---

## Round 3 · Author Response

Dear Editor,

We thank the referees for their careful reading of our manuscript and for their positive assessment. Here are our responses to their questions/comments.

Reply to Referee A:

(1)"In Fig. 3(c) the label should be J_{α,−λ}(t) (sign missing)."

Thank You. The figure was actually correct, there was a typo in the caption. We fixed it.

(2)"In Fig. 5(b) for the solid line there is only one trajectory to be seen. Do both particles indeed follow exactly the same trajectory? If yes is there a simple explanation why this happens?"

The particles corresponding to the solid lines in Fig.5(b) don’t follow exactly the same trajectory, but very close ones. Since lambda~pi/2, they are initially almost still, and their velocities remain very close to one another also when they enter the light cone. This can be understood using Eq. 14 and noting that T_{alpha,beta,-pi/2-mu}=T_{alpha,beta,pi/2-mu}, a_{alpha,-\pi/2}=a_{alpha,-pi/2}, v_{alpha,\pm pi/2}=0.

We added a sentence in the caption of Fig.5 to clarify this point.

(3)"What would change if the group velocity did not have only a single maximum?"

If the group velocity in one of the two leads does not have a single maximum we are unable to prove the non-crossing condition for particles with opposite rapidities. In particular, it is not generically possible to write \Phi_{\alpha,\lambda}(\zeta)-\Phi_{\alpha,-\lambda}(\zeta) in terms of an integral of a positive function, as done in Eq. 113.

Reply to Referee B:

"The only small improvement I would suggest is in the explanations around eq. 12. For appropriate context, I would suggest to mention that this is the hydrodynamic approximation. It is also important to mention that the result holds for x scaling with t as well, as such a scaling is what is used throughout the paper (and maybe mention that on the right-hand side, the only x,t dependence is in the density matrix; the x in the operator is not relevant)."

We agree with the referee: Eq. 12 holds in the hydrodynamic regime. Under very mild assumptions on the initial state this regime is reached for large times after the quench, when the system reached local equilibrium. This is the meaning of the large time limit in Eq. 12. In particular, this is expected to happen for very general inhomogeneous quench protocols, not only the bipartite quench considered in our work. For these reasons we decided to keep the discussion in Sec.3 at the general level, specifying the results to the case under exam in Sec.3.1.

We have added a sentence below Eq. 12 to clarify this point.

Regarding the x dependence in the operator, this can be removed when the state \rho_s(x,t) is homogeneous for each x and t. After Eq. 12 we argue that this is the case for integrable models, but to make the point clearer we decided to split the reasoning in two steps: first we argue that \rho_s(x,t) exists and then that is homogeneous at every fixed x and t.

---

## Round 3 · List of Changes

- Typo corrected in caption of Fig. 3

- Sentence added after Eq. 12

- Sentence added in caption of Fig. 5

- Other minor typos corrected throughout the manuscript

You are currently on this page

Resubmission 1903.00467v3 on 13 June 2019

---

## Editorial Decision

published